# MODEM: A Morton-Order Degradation Estimation Mechanism for Adverse Weather Image Recovery

**Hainuo Wang, Qiming Hu, Xiaojie Guo**[*]

College of Intelligence and Computing, Tianjin University, Tianjin 300350, China
hainuo@tju.edu.cn huqiming@tju.edu.cn xj.max.guo@gmail.com

## Abstract

Restoring images degraded by adverse weather remains a significant challenge due to the highly non-uniform and spatially heterogeneous nature of weather-induced artifacts, *e.g.*, fine-grained rain streaks versus widespread haze. Accurately estimating the underlying degradation can intuitively provide restoration models with more targeted and effective guidance, enabling adaptive processing strategies. To this end, we propose a Morton-Order Degradation Estimation Mechanism (MODEM) for adverse weather image restoration. Central to MODEM is the Morton-Order 2D-Selective-Scan Module (MOS2D), which integrates Morton-coded spatial ordering with selective state-space models to capture long-range dependencies while preserving local structural coherence. Complementing MOS2D, we introduce a Dual Degradation Estimation Module (DDEM) that disentangles and estimates both global and local degradation priors. These priors dynamically condition the MOS2D modules, facilitating adaptive and context-aware restoration. Extensive experiments and ablation studies demonstrate that MODEM achieves state-of-the-art results across multiple benchmarks and weather types, highlighting its effectiveness in modeling complex degradation dynamics. Our code will be released at here.

## 1 Introduction

Computer vision systems are increasingly integral to critical applications, such as autonomous driving [51, 1] and intelligent surveillance [49, 70], demanding reliable performance in diverse environments. However, their effectiveness deteriorates significantly under adverse weather conditions, such as rain [16, 42, 57, 75, 78, 92], haze [76, 77, 21, 62, 71, 83, 85], and snow [38, 45, 59, 87, 13, 12], which introduce complex visual artifacts and obscure critical scene information. Thus, effectively restoring clear images from such weather-degraded inputs is essential to boost the robustness and safety of modern computer vision technologies in real-world deployments.

Early task-specific methods [2, 4, 23, 60, 61, 30, 32, 90] largely rely on physical models or hand-crafted statistical priors tailored to individual weather phenomena. Due to the limited feature representation capabilities, these schemes are brittle in the face of complex scenes or deviations from assumed degradation patterns. With the advent of deep learning, numerous deep networks have achieved impressive performance by training on large-scale datasets for specific tasks such as image deraining [16, 42, 57, 75, 78, 5, 41, 69, 83], dehazing [76, 77, 21, 62, 71, 83, 85], or desnowing [38, 45, 59, 87]. These models excel at implicitly learning the inverse mapping through extensive supervision. But their highly specialized nature necessitates separate models for each weather condition, severely limiting scalability and practical deployment in unconstrained environments.

Recently, much attention has been directed toward unified or multi-task frameworks designed to cope with various weather conditions within a single model [95, 67, 11, 40, 55, 65, 79]. These

---

[*]Corresponding Author

39th Conference on Neural Information Processing Systems (NeurIPS 2025).

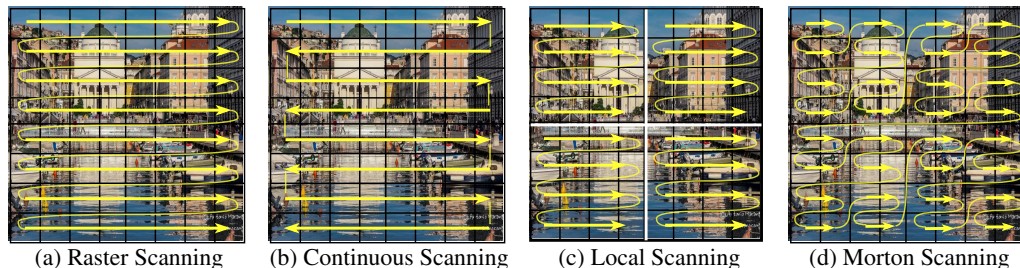

| (a) Raster Scanning | (b) Continuous Scanning | (c) Local Scanning | (d) Morton Scanning |

Figure 1: Comparison of various image scanning methods. (a) Raster Scanning. (b) Continuous Scanning. (c) Local Scanning. (d) Morton Scanning, which can better preserve spatial locality of neighboring pixels in the resulting 1D sequence, beneficial for capturing contextual information.

approaches adopt a variety of sophisticated strategies, including architecture search for optimal generalization [40], disentanglement of weather-shared and weather-specific representations [95], knowledge distillation and regularization [11], generative diffusion modeling [55], learned priors or codebooks [79], and transformer-based architectures that offer strong global modeling capabilities [67, 65]. Although these models mark notable progress and offer broader applicability, they still struggle with the challenge of modeling the inherently distinct and often highly spatially heterogeneous characteristics of different weather degradations. For example, haze tends to cause smooth intensity attenuation, whereas rain streaks and snowflakes introduce sharp, local occlusions with specific structural patterns. Most existing architectures lack dedicated degradation estimation mechanisms to explicitly model and leverage these fine-grained, spatially variant degradation patterns, which undermines their performance in real-world deployment. Therefore, *effective **estimation** of spatially variant degradation characteristics, encompassing both global trends and local structures, as **guidance** is critical for adaptive and context-aware restoration under dynamic weather conditions.*

For an element (*e.g.*, pixel) of a degraded image, its degradation characteristic can be regarded as a latent "state" within a degradation space, encapsulating the influence of adverse weather on its visual appearance. These states evolve spatially, governed by both local (*e.g.*, rain streaks) and non-local patterns (*e.g.*, drifting fog). This perspective naturally lends itself to State Space Modeling (SSM) [17]. The connection between the image degradation modeling and SSM is formally given in Sec. 3. By treating degradation as a structured sequence of evolving states, SSM can capture long-range contextual dependencies while maintaining sensitivity to local features. When coupled with degradation estimation mechanisms, SSM empowers the restoration process to be contextually aware and spatially adaptive, effectively aligning restoration strategies with the heterogeneous and scene-specific nature of weather-induced artifacts.

Motivated by the above insight, we propose a novel Morton-Order Degradation Estimation Mechanism (MODEM) that **estimates** degradation state via SSM and **guides** adaptive restoration in degraded images. At the heart of MODEM lies the Morton-Order 2D-Selective-Scan Module (MOS2D), which integrates SSM with Morton-coded spatial traversal. Unlike conventional raster scanning, the Morton-order scan follows a hierarchical, locality-preserving sequence [50, 34, 53, 31, 7, 54] that enables structured and efficient modeling of both local and long-range dependencies, as shown in Fig. 1(d). To complement MOS2D, we further introduce a Dual Degradation Estimation Module (DDEM), designed to extract two complementary forms of degradation information from the input: (i) a global degradation representation that encapsulates high-level weather characteristics such as type and severity, and (ii) a set of spatially adaptive kernels that encode local degradation structures and variations. These dual degradation representations are then utilized to dynamically modulate the restoration process. Specifically, the global representation adaptively influences feature transformations within MODEM, while the spatially adaptive kernels guide the matrix construction within MOS2D, refining the spatial dependencies captured along the Morton-order sequence. This dual-modulation strategy equips the restoration network with both global awareness and local sensitivity, thereby delivering more precise and effective restoration. Our primary contributions can be summarized as follows:

- We propose a novel Morton-Order Degradation Estimation Mechanism that introduces the MOS2D module, which integrates Morton-coded spatial ordering with selective state space modeling to effectively capture spatially heterogeneous weather degradation dynamics.
- We design a Dual Degradation Estimation Module that jointly estimates global degradation descriptors and spatially adaptive kernels. These two complementary representations are used to dynamically modulate MOS2D, allowing contextually aware and spatially adaptive restoration.

- Extensive experiments and ablation studies are conducted to demonstrate the effectiveness of the proposed MODEM, and its superiority over other state-of-the-art competitors in restoring images under diverse and complex adverse weather conditions.

## 2 Related Work

This section briefly reviews representative approaches to single adverse weather restoration including rain streak removal, raindrop removal, haze removal and snow removal, as well as unified all-in-one models. Additionally, recent advances in SSM-based image restoration techniques are also discussed.

*Rain Streak Removal.* Traditional rain streak removal methods typically relied on image decomposition [32] or tensor-based priors [30]. With the advent of deep learning, various models have emerged, like the deep detail network for splitting rain details [16], recurrent networks for context aggregation in single images [41], and spatio-temporal aggregation in videos [42]. Further developments include uncertainty-guided multi-scale designs [78], density-aware architectures [83], spatial attention [69], joint detection-removal frameworks [75], and NAS-based attentive schemes [5].

*Raindrop Removal.* Early efforts addressed specific scenarios [15] or adherent raindrops in videos [80]. Deep learning subsequently provided more generalizable solutions, including learning from synthetic photorealistic data [22], using attention-guided GANs for realistic inpainting [57], and dedicated networks for visibility through raindrop-covered windows [58]. General restoration architectures like Dual Residual Networks [43] and Adaptive Sparse Transformers [92] are also applicable, with advanced dual attention-in-attention models [86] tackling joint rain streak and raindrop removal.

*Haze Removal.* Traditional image dehazing methods often utilized statistical priors the Dark Channel Prior (DCP) [23], non-local similarity techniques [4], or multi-scale fusion [2] But these approaches faced robustness issues in diverse hazy scenes due to strong underlying assumptions. Deep learning has since become dominant, with methods focusing on estimating transmission and atmospheric light, sometimes using depth awareness [76] or unpaired learning via decomposition [77]. Other approaches tackle haze density variations [83, 85], domain adaptation [62], and contrastive learning [71].

*Snow Removal.* While specific traditional priors for snow removal are less common compared to other weather conditions, principles from general image restoration were often adapted. Deep learning solutions have gained traction, including powerful general-purpose backbones [13, 12] and more specialized networks. Desnowing-specific models often involve context-aware designs [45] or leverage semantic and depth priors [87]. Techniques inspired by classical matrix decomposition [59] and methods for online video processing [38] have also been integrated into deep frameworks.

While these specialized models excel under specific weather conditions, their limited generalization requires separate models for each degradation type, hindering real-world practicality.

*Unified Adverse Weather Restoration.* To mitigate the inefficiency of deploying multiple models for complex weather conditions, unified models have been devised [95, 67, 11, 40, 55, 65, 79]. However, these unified ones face a fundamental challenge that *the degradation space covered by unified models is exponentially larger than that of weather-specific counterparts*, substantially increasing the complexity of accurate modeling and generalization. To tackle this, existing approaches employ different strategies to learn more generalizable representations. Early efforts included using NAS to find a unified structure [40] and leveraging Transformers like TransWeather [67] for their global context modeling. Others focused on training strategies, such as two-stage knowledge learning with multi-contrastive regularization [11]. More recent works have explored disentangling weather-general and weather-specific features [95]), adapting powerful generative models like diffusion models [55]), incorporating learned codebook priors [79]), and enhancing Transformers with global image statistics like Histoformer [65]. These methods showcase a trend towards more sophisticated and adaptive unified restoration. Despite progress, the unified models still struggle to effectively capture the diverse and spatially heterogeneous characteristics of different weather phenomena.

*SSM-based Image Restoration.* Recent State Space Models (SSMs) [18, 17], notably Mamba [17] with its selective scan mechanism, offer efficient long-sequence modeling and have rapidly permeated computer vision [52, 94, 44, 48, 64]. Inspired by these advancements, SSMs are increasingly applied to image restoration. General frameworks like MambaIR [20], VMambaIR [63], and CU-Mamba [14] demonstrate the potential of Mamba-based models as strong baselines. Specific low-level vision applications have also seen tailored solutions. For example, WaterMamba [19] has been developed

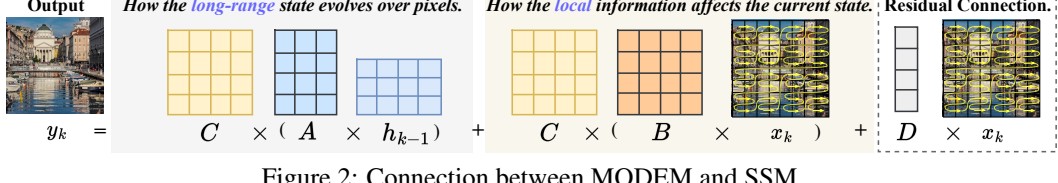

Figure 2: Connection between MODEM and SSM.

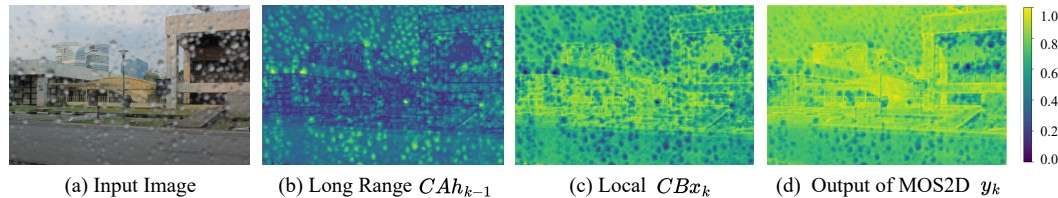

(a) Input Image     (b) Long Range $CAh_{k-1}$     (c) Local $CBx_k$     (d) Output of MOS2D $y_k$

Figure 3: With respect to a sample (a), (b)-(d) visualize the long-range $CAh_{k-1}$, (c) local $CBx_k$, and (d) output of MOS2D $y_k$, respectively. More cases can be found in Appendix A.4

for underwater image enhancement, and IRSRMamba [26] addresses infrared super-resolution. Low-light image enhancement has been tackled by models like RetinexMamba [3] and LLEMamba [89], while efficient SSM designs have been proposed for image deblurring [35]. In image deraining, notable works include FourierMamba [37] and FreqMamba [91]. These efforts underscore the growing success of SSMs in addressing image restoration challenges. Our work builds upon this trend, leveraging SSMs for adaptive modeling of spatially heterogeneous weather degradations.

## 3 Bridging Image Degradation Estimation and State Space Modeling

Restoring images corrupted by adverse weather is inherently a highly ill-posed problem. In general, the relationship between an observed degraded image $x$ and its clean version $y$ can be modeled as:

$$x = \text{Degrade}(y, \theta := \{\theta_G, \theta_L\}) \quad \Leftrightarrow \quad y = \text{Restore}(x, \theta := \{\theta_G, \theta_L\}), \tag{1}$$

where $\text{Degrade}(\cdot, \theta)$ designates a complex, often spatially varying weather degradation function parameterized by $\theta$. By considering the spatial influence of degradation, the parameter set ($\theta$) can be partitioned into long-range/global degradation ($\theta_G$, *e.g.* atmospheric haze) and local one ($\theta_L$, *e.g.* rain streak). The objective is to recover the latent clean $y$ from the degraded observation $x$. The core challenge lies in the fact that both the degradation process and the artifacts are unknown and can vary significantly across scenes and weather types. Thus, estimating the degradation characteristic $\theta$ and constructing the function $\text{Restore}(\cdot, \theta)$ in Eq. (1), is central to solving the problem.

We propose that State Space Models (SSMs) [17] offer a suitable and robust framework for this degradation estimation task. Recall the core recurrence of a discrete-time SSM:

$$h_k = Ah_{k-1} + Bx_k, \quad y_k = Ch_k + Dx_k, \tag{2}$$

where $x_k$ is the input Morton-order sequence at step $k$, $h_k$ is the latent hidden state summarizing historical context, and $y_k$ is the output. The matrices $A$, $B$, and $C$ are learnable parameters that define the SSM's dynamics and output mapping. Please notice that the term $Dx_k$ is a practical (non-theoretical) addition for training ease, which is analogous to residual connection. To better understand how the state dynamics contribute to the output $y_k$, we can analyze the system by omitting the residual $Dx_k$ and substitute $h_k$ into the $y_k = Ch_k$ part, as follows:

$$y_k = CAh_{k-1} + CBx_k. \tag{3}$$

By comparing Eq. (1) with Eq. (3), we interpret SSM components as degradation estimators, as illustrated in Fig. 2. The term $CAh_{k-1}$ conceptually models the long-range, spatially propagated degradation context, as shown in Fig. 3(b). The previous hidden state $h_{k-1}$ summarizes accumulated degradation cues (*e.g.*, overall haze level, rain intensity) along the Morton-order scan path. The state transition matrix $A$ then models how these summarized characteristics evolve or persist as the scan progresses spatially (*e.g.*, the smooth attenuation of visibility due to widespread haze or the consistent statistical properties of rain streaks over a larger area.). It essentially dictates "how the long-range state evolves over pixels" as noted in Fig. 2. The output matrix $C$ then translates this evolved context $Ah_{k-1}$ into a contribution to the current output $y_k$, reflecting broader, non-local degradation patterns.

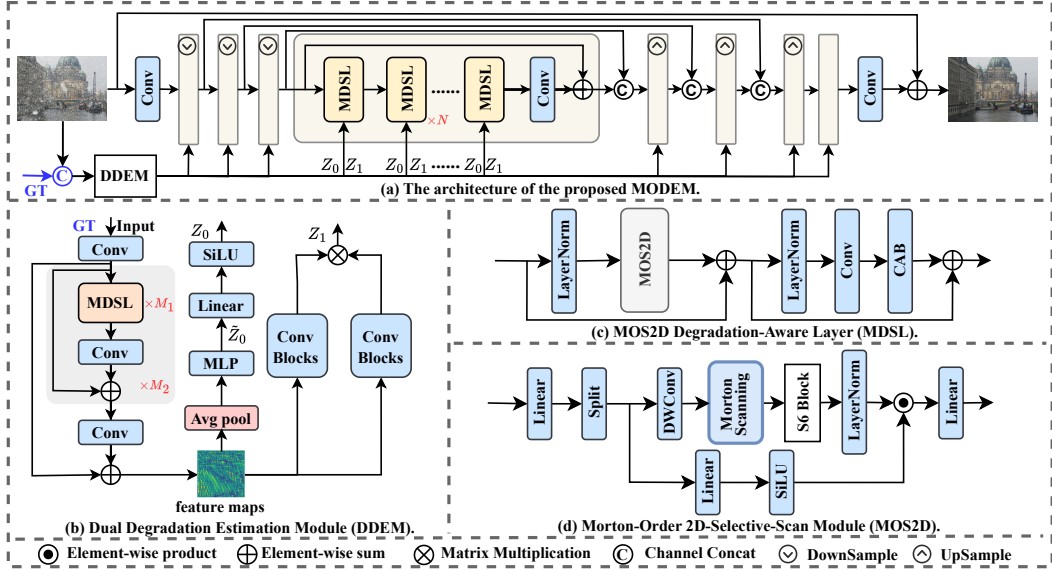

Figure 4: (a) Overall architecture of MODEM. (b) The DDEM for extracting global descriptor $Z_0$ and adaptive degradation kernel $Z_1$ degradation priors. (c) The MDSL incorporating the core MOS2D module (d) within a residual block. The **blue-colored** components indicate elements exclusive to the first training stage. $N$, $M_1$, $M_2$ denote the number of the corresponding module, respectively.

Concurrently, the $CBx_k$ term accounts for the impact of the immediate, local input features $x_k$, as visualized in Fig. 3(c). The input matrix $B$ determines "how the local information affects the current state" (Fig. 2), allowing features from $x_k$ (*e.g.*, local degradation artifacts like dense fog patches, or crucial local image content like fine textures/sharp edges) to directly influence the current hidden state $h_k$. This enables the SSM to react to fine-grained local evidence, with $C$ projecting this locally-informed component into $y_k$ for specific adjustments based on immediate observations.

## 4 Morton-Order Degradation Estimation Mechanism

The Morton-Order Degradation Estimation Mechanism (MODEM), depicted in Fig. 4(a), employs a two-stage training strategy designed to accurately learn degradation characteristics.

*Stage 1:* In the first stage, the Dual Degradation Estimation Module (DDEM), shown in Fig. 4(b), is provided with both the degraded image $I_{\text{LQ}}$ and its corresponding ground-truth $I_{\text{GT}}$ from the training set. This allows the DDEM to explicitly learn the mapping from an image pair to its underlying degradation patterns. It outputs two degradation priors: a global descriptor $Z_0$ and a spatially adaptive degradation kernel $Z_1$. These priors, representing the degradation information, are then injected into MODEM's main restoration backbone. The backbone itself, comprised of $N$ stacked MOS2D Degradation-Aware Layers (MDSLs) detailed in Fig. 4(c), only receives the degraded image $I_{\text{LQ}}$ and uses these priors to perform adaptive restoration. Each MDSL leverages the Morton-Order 2D-Selective-Scan module (MOS2D), shown in Fig. 4(d) and Fig. 5, to adaptively modulate features based on the degradation priors $Z_0$ and $Z_1$. The degradation representation learned by the DDEM in this stage serves as the supervisory target for the Stage 2.

*Stage 2:* In the second stage, the inputs to both the DDEM and the main backbone consist of only the degraded image $I_{\text{LQ}}$, without any GT. The Stage 2 model inherits its parameters from Stage 1, and the DDEM in this stage receives only the degraded image $I_{\text{LQ}}$ as input. Simultaneously, there is a frozen DDEM receiving both the GT and the degraded image. The degradation information from this frozen DDEM is then used to supervise the trainable DDEM.

### 4.1 Dual Degradation Estimation Module (DDEM)

The Dual Degradation Estimation Module (DDEM), shown in Fig. 4(b), extracts global degradation descriptor $Z_0$ and adaptive degradation kernel $Z_1$. In the first stage, DDEM processes the degraded image $I_{\text{LQ}}$ and ground-truth $I_{\text{GT}}$. Otherwise, it uses only $I_{\text{LQ}}$. The input undergoes several

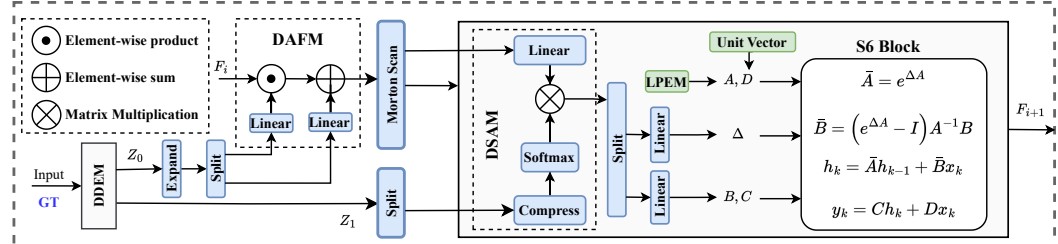

Figure 5: Detailed illustration of the degradation modulation mechanism within a MOS2D module in the main restoration backbone, which employs the Degradation-Adaptive Feature Modulation (DAFM) and Degradation-Selective Attention Modulation (DSAM) to dynamically adjust feature representations based on the degradation priors $Z_0$ and $Z_1$.

MOS2D Degradation-Aware Layers (MDSLs) to extract degradation feature map $F \in \mathbb{R}^{H \times W \times C_d}$. Subsequently, the global descriptor $Z_0$ and adaptive kernels $Z_1$ are derived from $F$ as follows:

$$\tilde{Z} = \text{MLP}(\text{AvgPool}(F)), \quad Z_0 = \sigma(\text{Linear}(\tilde{Z})), \quad Z_1 = \text{Conv}(F) \times \text{Conv}(F)^T, \quad (4)$$

where $\sigma(\cdot)$ denotes the SiLU activation function, $\tilde{Z} \in \mathbb{R}^{4C_d}$ will be used for degradation supervision. These priors $Z_0 \in \mathbb{R}^{C_d}$ and $Z_1 \in \mathbb{R}^{C_{d1} \times C_{d2}}$ then guide the main restoration network.

*MOS2D Degradation-Aware Layers (MDSL).* The iterative application of MDSL enhances sensitivity to degradation patterns, yielding enriched feature maps. For the feature map $F_i$ at the $i$-th MDSL, this process of MDSL can be formulated as:

$$\tilde{F}_i = \text{MOS2D}(\text{LN}(F_i)) + F_i, \quad F_{i+1} = \text{CAB}(\text{Conv}(\text{LN}(\tilde{F}_i))) + \tilde{F}_i, \quad (5)$$

where $\text{CAB}(\cdot)$ denotes the Channel Attention Block. $\text{LN}(\cdot)$ represents Layer Normalization.

## 4.2 Morton-Order 2D-Selective-Scan Module (MOS2D)

To effectively model spatially heterogeneous degradations in 2D images, our MOS2D employs Morton-Order scan, as illustrated in Fig. 1(d). This converts 2D spatial features into locality-preserving 1D sequence, facilitating structured feature interaction and aggregation by SSM.

Specifically, the Morton encoding $z$ maps 2D pixel coordinates $(i, j)$, where $0 \le i < H, 0 \le j < W$, to a 1D index by interleaving the bits of their binary representations. For $i = (i_n, \ldots, i_0)_2$ and $j = (j_n, \ldots, j_0)_2$, with $n = \lceil \log_2(\max(H, W)) \rceil - 1$, the encoding $z$ is:

$$z = \text{interleave}(i, j) = (j_n, i_n, j_{n-1}, i_{n-1}, \ldots, j_1, i_1, j_0, i_0)_2. \quad (6)$$

In the DDEM, Morton-order coding is followed by standard SSM operations to effectively extract global degradation descriptor $Z_0$ and adaptive degradation kernel $Z_1$. In contrast, in our main restoration backbone, the MOS2D module employs degradation-aware modulations using $Z_0$ and $Z_1$ through *DAFM* and *DSAM*, detailed in Fig. 5. This dual-modulation ensures that the MOS2D is dynamically conditioned on both long range contextually aware and spatially adaptive restoration.

*Degradation-Adaptive Feature Modulation (DAFM).* The $i$-th layer's input feature map $F_i$ is first modulated by the global degradation descriptor $Z_0$. As shown in Fig. 5, $Z_0$ is expanded and split to produce channel-wise adaptive weights $Z_0^w$ and biases $Z_0^b$, which applied to $F_i$ using a feature-wise linear modulation operation, thus incorporates global degradation characteristics:

$$F_{\text{DAFM}} = (Z_0^w \odot F_i) + Z_0^b, \quad Z_0^w, Z_0^b = \text{Split}(\text{Linear}(Z_0)), \quad (7)$$

where $\odot$ denotes element-wise multiplication.

*Degradation-Selective Attention Modulation (DSAM).* To further guide the S6 Block with local degradation information, the spatially adaptive kernel $Z_1$ is utilized as follows:

$$F_{\text{DSAM}} = W_F F_{\text{DAFM}} \times \text{Softmax}(W_Z Z_1), \quad (8)$$

where $W_F$ and $W_Z$ are learnable linear projection matrices.

*Degradation-Guided S6 Block.* The core selective scan operation is performed by our Degradation-Guided S6 Block. Its parameters are dynamically adapted based on the degradation-sensitive features

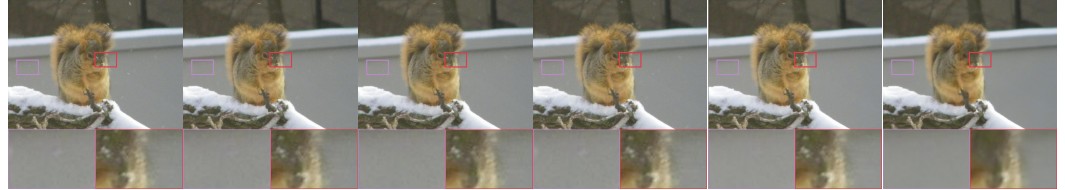

| (a) Input | (b) WGWSNet | (c) WeatherDiff | (d) Histoformer | (e) Histoformer† | (f) Ours |

Figure 6: Visual comparison on the real world snow dataset [45]. Compared to the prior methods [95, 55, 65], where "†" denotes the Histoformer [65] for real snow from their official repository, our MODEM achieves superior results without additional training.

Table 1: Quantitative comparison with recent state-of-the-art unified methods [40, 67, 11, 95, 55, 79, 65] across various datasets. The best and second-best results are in **bold** and underlined, respectively.

| Methods | Snow100K-S | | Snow100K-L | | Outdoor | | RainDrop | | Average | |
|---|---|---|---|---|---|---|---|---|---|---|
| | PSNR | SSIM | PSNR | SSIM | PSNR | SSIM | PSNR | SSIM | PSNR | SSIM |
| All-in-One [40] | - | - | 28.33 | 0.8820 | 24.71 | 0.8980 | 31.12 | 0.9268 | 28.05 | 0.9023 |
| TransWeather [67] | 32.51 | 0.9341 | 29.31 | 0.8879 | 28.83 | 0.9000 | 30.17 | 0.9157 | 30.21 | 0.9094 |
| Chen *et al.* [11] | 34.42 | 0.9469 | 30.22 | 0.9071 | 29.27 | 0.9147 | 31.81 | 0.9309 | 31.43 | 0.9249 |
| WGWSNet [95] | 34.31 | 0.9460 | 30.16 | 0.9007 | 29.32 | 0.9207 | 32.38 | 0.9378 | 31.54 | 0.9263 |
| WeatherDiff$_{64}$ [55] | 35.83 | 0.9566 | 30.09 | 0.9041 | 29.64 | 0.9312 | 30.71 | 0.9312 | 31.57 | 0.9308 |
| WeatherDiff$_{128}$ [55] | 35.02 | 0.9516 | 29.58 | 0.8941 | 29.72 | 0.9216 | 29.66 | 0.9225 | 31.00 | 0.9225 |
| AWRCP [79] | 36.92 | 0.9652 | 31.92 | **0.9341** | 31.39 | 0.9329 | 31.93 | 0.9314 | 33.04 | 0.9409 |
| Histoformer [65] | 37.41 | 0.9656 | 32.16 | 0.9261 | 32.08 | 0.9389 | **33.06** | **0.9441** | 33.68 | 0.9437 |
| MODEM (Ours) | **38.08** | **0.9673** | **32.52** | 0.9292 | **33.10** | **0.9410** | 33.01 | 0.9434 | **34.18** | **0.9452** |

$F_{\text{DSAM}}$ derived from the DSAM stage. $F_{\text{DSAM}}$ is split into three components, which are then linearly transformed to generate the SSM parameters $B$, $C$, and $\Delta$:

$$F_\Delta, F_B, F_C = \text{Split}(F_{\text{DSAM}}), \quad \Delta = W_\Delta F_\Delta, \quad B = W_B F_B, \quad C = W_C F_C, \qquad (9)$$

where $W_\Delta$, $W_B$, and $W_C$ are learnable linear mappings. We use zero-order hold (ZOH) discretization with timescale $\Delta$ to obtain the discrete matrices $\bar{A} = e^{\Delta A}$ and $\bar{B} = (\exp(\Delta A) - I)A^{-1}B$, where $I$ is the identity matrix. The S6 Block then processes the Morton-Ordered sequence $x_k$, which is the Morton-ordered sequence derived from the DAFM features $F_{\text{DAFM}}$, as follows:

$$y_k = Ch_k + Dx_k, \quad h_k = \bar{A}h_{k-1} + \bar{B}x_k, \quad x_k = F_{\text{DAFM}}[z], \qquad (10)$$

where $h_k$ is the hidden state at step $k$. The dynamically generated $B$ and $C$ (and $\Delta$ influencing $\bar{A}, \bar{B}$) ensure that the state evolution and output generation are adaptive to the degradation characteristics.

## 4.3 Loss Function

Our model is trained in two stages. In the first stage, the loss $\mathcal{L}_{\text{stage1}}$ combines a $\mathcal{L}_1$ loss and a correlation loss $\mathcal{L}_{\text{cor}}$ [65] to ensure accuracy and structural fidelity between the output of MODEM $I_{\text{HQ}}$ and ground-truth $I_{\text{GT}}$:

$$\mathcal{L}_{\text{stage1}} = \mathcal{L}_1(I_{\text{HQ}}, I_{\text{GT}}) + \mathcal{L}_{\text{cor}}(I_{\text{HQ}}, I_{\text{GT}}), \qquad (11)$$

The correlation loss $\mathcal{L}_{\text{cor}}(I_{\text{HQ}}, I_{\text{GT}})$ [65] is based on the Pearson correlation coefficient $\rho(I_{\text{HQ}}, I_{\text{GT}})$:

$$\mathcal{L}_{\text{cor}}(I_{\text{HQ}}, I_{\text{GT}}) = \frac{1}{2}\left(1 - \rho(I_{\text{HQ}}, I_{\text{GT}})\right), \quad \rho(I_{\text{HQ}}, I_{\text{GT}}) = \frac{\sum_{i=1}^{N}(I_{i,\text{HQ}} - \bar{I}_{\text{HQ}})(I_{i,\text{GT}} - \bar{I}_{\text{GT}})}{N \cdot \sigma(I_{\text{HQ}}) \cdot \sigma(I_{\text{GT}})}. \quad (12)$$

where $N$ is the total number of pixels, $\bar{I}$ denotes mean, and $\sigma(I)$ denotes standard deviation.

In the second stage, we introduce a KL divergence loss $\mathcal{L}_{\text{KL}}$ to maintain consistency of the degradation representation $\tilde{Z}$ across stages. Let $\tilde{Z}_{0,\text{st1}}$ and $\tilde{Z}_{0,\text{st2}}$ be the $\tilde{Z}$ vectors from the first (fixed) and second stages, respectively. The KL divergence is computed between their softmax distributions $\phi(\cdot)$:

$$\mathcal{L}_{\text{KL}} = D_{\text{KL}}\left(\phi(\tilde{Z}_{0,\text{st1}}) \,\|\, \phi(\tilde{Z}_{0,\text{st2}})\right) = \sum_{j=1}^{4C_d} \phi(\tilde{Z}_{0,\text{st1}}(j)) \log\left(\frac{\phi(\tilde{Z}_{0,\text{st1}}(j))}{\phi(\tilde{Z}_{0,\text{st2}}(j))}\right). \qquad (13)$$

<table>
<tr><td colspan="5" align="center">Table 2: Desnowing Task</td></tr>
<tr><td rowspan="2">Methods</td><td colspan="2">Snow100K-S</td><td colspan="2">Snow100K-L</td></tr>
<tr><td>PSNR</td><td>SSIM</td><td>PSNR</td><td>SSIM</td></tr>
<tr><td>SPANet [69]</td><td>29.92</td><td>0.8260</td><td>23.70</td><td>0.7930</td></tr>
<tr><td>JSTASR [10]</td><td>31.40</td><td>0.9012</td><td>25.32</td><td>0.8076</td></tr>
<tr><td>RESCAN [41]</td><td>31.51</td><td>0.9032</td><td>26.08</td><td>0.8108</td></tr>
<tr><td>DesnowNet [45]</td><td>32.33</td><td>0.9500</td><td>27.17</td><td>0.8983</td></tr>
<tr><td>DDMSNet [87]</td><td>34.34</td><td>0.9445</td><td>28.85</td><td>0.8772</td></tr>
<tr><td>Restormer [82]</td><td>36.01</td><td>0.9579</td><td>30.36</td><td>0.9068</td></tr>
<tr><td>ConvIR [13]</td><td>37.98</td><td>0.9686</td><td>32.11</td><td>**0.9300**</td></tr>
<tr><td>FSNet [12]</td><td>37.42</td><td>0.9654</td><td>31.62</td><td>0.9246</td></tr>
<tr><td>MODEM (Ours)</td><td>**38.08**</td><td>**0.9673**</td><td>**32.52**</td><td>0.9292</td></tr>
</table>

<table>
<tr><td colspan="3" align="center">Table 3: Raindrop Removal Task</td></tr>
<tr><td rowspan="2">Methods</td><td colspan="2">RainDrop</td></tr>
<tr><td>PSNR</td><td>SSIM</td></tr>
<tr><td>pix2pix [27]</td><td>28.02</td><td>0.8547</td></tr>
<tr><td>DuRN [43]</td><td>31.24</td><td>0.9259</td></tr>
<tr><td>RaindropAttn [58]</td><td>31.44</td><td>0.9263</td></tr>
<tr><td>AttentiveGAN [57]</td><td>31.59</td><td>0.9170</td></tr>
<tr><td>IDT [73]</td><td>31.87</td><td>0.9313</td></tr>
<tr><td>MAXIM [66]</td><td>31.87</td><td>0.9352</td></tr>
<tr><td>Restormer [82]</td><td>32.18</td><td>0.9408</td></tr>
<tr><td>AST [92]</td><td>30.57</td><td>0.9333</td></tr>
<tr><td>MODEM (Ours)</td><td>**33.01**</td><td>**0.9434**</td></tr>
</table>

<table>
<tr><td colspan="3" align="center">Table 4: Deraining & Dehazing Task</td></tr>
<tr><td rowspan="2">Methods</td><td colspan="2">Outdoor-Rain</td></tr>
<tr><td>PSNR</td><td>SSIM</td></tr>
<tr><td>CycleGAN [93]</td><td>17.62</td><td>0.6560</td></tr>
<tr><td>pix2pix [27]</td><td>19.09</td><td>0.7100</td></tr>
<tr><td>HRGAN [39]</td><td>21.56</td><td>0.8550</td></tr>
<tr><td>PCNet [29]</td><td>26.19</td><td>0.9015</td></tr>
<tr><td>MPRNet [81]</td><td>28.03</td><td>0.9192</td></tr>
<tr><td>NAFNett [9]</td><td>29.59</td><td>0.9027</td></tr>
<tr><td>Restormer [82]</td><td>30.03</td><td>0.9215</td></tr>
<tr><td>MODEM (Ours)</td><td>**33.10**</td><td>**0.9410**</td></tr>
</table>

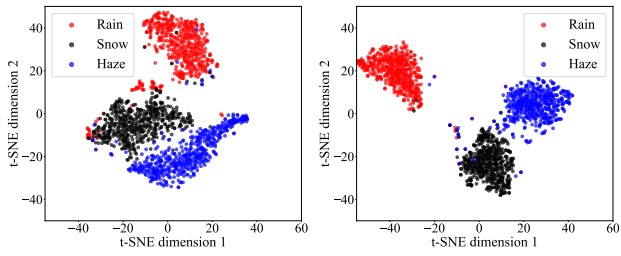

(g) T-SNE of Histoformer [65]      (h) T-SNE of MODEM

Figure 7: T-SNE results of Histoformer [65] and MODEM, which reflect that MODEM exhibits better clustering of features corresponding to different weather types than Histoformer [65].

The total loss for the second stage, $\mathcal{L}_{\text{stage2}}$, can be formulated as:

$$\mathcal{L}_{\text{stage2}} = \mathcal{L}_1(I_{\text{HQ}}, I_{\text{GT}}) + \mathcal{L}_{\text{cor}}(I_{\text{HQ}}, I_{\text{GT}}) + \mathcal{L}_{\text{KL}}(\tilde{Z}_{0,\text{st1}}, \tilde{Z}_{0,\text{st2}}). \tag{14}$$

## 5 Experimental Validation

Our MODEM is implemented in PyTorch [56] and trained on 4 NVIDIA RTX 3090 GPUs in two stages. We employ the AdamW optimizer [47] with a Cosine Annealing Restart Cyclic LR scheduler [46]. It is trained on an all-weather dataset [67, 65], consistent with prior works [40, 67, 55, 65, 95, 79]. For evaluation, MODEM is tested on established benchmarks including Test1 [39, 40], RainDrop [57], Snow100K-L/S [45], and a real-world snow test set [45]. Due to the page limit, more details are given in Appendix A.1 and A.2.

### 5.1 Comparisons with State-of-the-arts

*Quantitative Comparison.* Our quantitative evaluation assesses MODEM against both unified models [40, 67, 11, 95, 55, 79, 65] and task-specific methods [69, 10, 41, 45, 87, 9, 82, 43, 27, 58, 57, 73, 66, 93, 39, 29, 81]. As reported in Table 1, MODEM achieves SOTA performance, with an average PSNR improvement of 0.5dB across benchmarks [39, 40, 57, 45], and a notable 1.02dB gain on Outdoor-Rain [39, 40]. While slightly below Histoformer on RainDrop [57] by 0.05dB PSNR, MODEM yields compelling visual results, a point further elaborated in our qualitative comparisons. Furthermore, comparisons with leading task-specific methods in Tables 2 to 4 confirm MODEM consistently attains SOTA results. These evaluations underscore MODEM's capability to effectively estimate and adapt to diverse, spatially heterogeneous weather degradations. Additionally, the comparisons of complexity can be found in Appendix A.3.

*Qualitative comparison.* We provide qualitative comparisons across diverse scenarios in Figs. 6 and 8. For image desnowing on Snow100K [45], MODEM effectively removes heavy snowflakes and visual artifacts that other models [95, 55, 65] struggle to address. For joint deraining and dehazing on Outdoor-Rain [39, 40], MODEM excels in restoring richer texture details and produces images with noticeably fewer artifacts. For raindrop removal on the Raindrop [57], MODEM again

Table 5: Comparison of perceptual metrics, including referenced (LPIPS↓) and non-referenced (Q-Align↑, MUSIQ↑) scores. Best results are **bolded**; second-best are underlined.

| | Method | Snow100K-L | Snow100K-S | Outdoor | Raindrop | Snow100K-Real |
|---|---|---|---|---|---|---|
| LPIPS | Histoformer [65] | 0.0919 | 0.0445 | 0.0778 | 0.0672 | – |
| | WeatherDiff [55] | 0.0982 | 0.0541 | 0.0887 | **0.0615** | – |
| | MODEM (Ours) | **0.0880** | **0.0407** | 0.0699 | 0.0650 | – |
| Q-Align | Histoformer [65] | 3.7207 | 3.7598 | 4.1445 | 4.0156 | 3.5449 |
| | WeatherDiff [55] | 3.4531 | 3.5293 | 3.8691 | 4.0000 | 3.4512 |
| | MODEM (Ours) | **3.7324** | **3.7695** | **4.1875** | **4.0664** | **3.5586** |
| MUSIQ | Histoformer [65] | **64.2526** | 64.2581 | 67.7461 | 68.4852 | 59.4040 |
| | WeatherDiff [55] | 62.6267 | 63.1278 | 67.4814 | 69.3608 | 59.4493 |
| | MODEM (Ours) | 64.2438 | **64.2853** | **68.2926** | **69.7925** | **59.6042** |

Table 6: Comparison of different methods on various real-world datasets using the Q-Align metric.

| Method | Snow100K-Real | RainDrop | NTURain | RESIDE | WeatherStream |
|---|---|---|---|---|---|
| WeatherDiff [55] | 3.4531 | 4.0000 | 3.2031 | **3.4219** | 1.9561 |
| Histoformer [65] | 3.7207 | 4.0156 | 3.2266 | 3.2891 | 1.9434 |
| MODEM (Ours) | **3.7324** | **4.0664** | **3.2891** | 3.3164 | **1.9863** |

demonstrates superior detail preservation and artifact reduction. Furthermore, on real-world snowy images [45], MODEM achieves superior results as shown in Fig. 6 even without any additional fine-tuning, showcasing excellent generalization and real-world applicability, which can be attributed to its profound understanding and adaptive modeling of degradation characteristics. Due to the page limit, more visual comparisons can be found in Appendix A.8.

## 5.2 Perceptual Quality and Real-World Performance

*Comparison of perceptual metrics.* We report referenced (LPIPS [88]) and non-referenced (Q-Align [72], MUSIQ [33]) scores, all computed using the pyiqa library [6], with the same settings applied for all methods. As shown in Table 5, our method achieves SOTA perceptual quality.

*Comparison of real-world datasets.* To provide quantitative evidence of our model's performance on the challenging real-world scenarios, we evaluated our model on the real-world data from the Snow100K-Real [45], RainDrop [57], RESIDE [36], NTURain [8] and WeatherStream [84] datasets using the Q-Align [72]. The results are presented in Table 6. As the results show, our method achieves state-of-the-art performance on real-world datasets when compared to the previous state-of-the-art method, Histoformer [65], and the diffusion-based approach, WeatherDiff [55].

## 5.3 Ablation Studies

We evaluate the impact of the Morton-Order scan, DDEM, DAFM and DSAM. Morton-Order scan facilitates structured spatial feature processing within the SSM, enabling the model to better capture contextual dependencies and preserve local structural coherence. As shown in Table 7, configurations incorporating the Morton scan generally yield improved performance, underscoring its benefit in organizing spatial information for sequential modeling. DDEM provides the degradation priors $Z_0$ and $Z_1$ that guide the DAFM and DSAM. As indicated in Table 7, configurations lacking DDEM exhibit a significant performance drop. DAFM, which utilizes the global degradation prior $Z_0$, plays a crucial role in adaptively modulating features based on the overall estimated weather type and severity. DSAM, guided by the spatially adaptive kernel $Z_1$, allows the MOS2D to selectively focus on and modulate features pertinent to local degradation characteristics or specific image regions. While removing DSAM shows marginal gains on certain snow metrics, it impairs performance on tasks like raindrop removal, which demands strong local adaptation due to the highly local nature of raindrops, and on mixed rain&haze scenarios where intricate local texture recovery is challenging. Thus, DSAM, guided by the spatially adaptive kernel $Z_1$, is crucial for achieving robust, balanced performance across diverse weather conditions by guiding the network to selectively address these local characteristics. The combination of all components employed in full MODEM, consistently

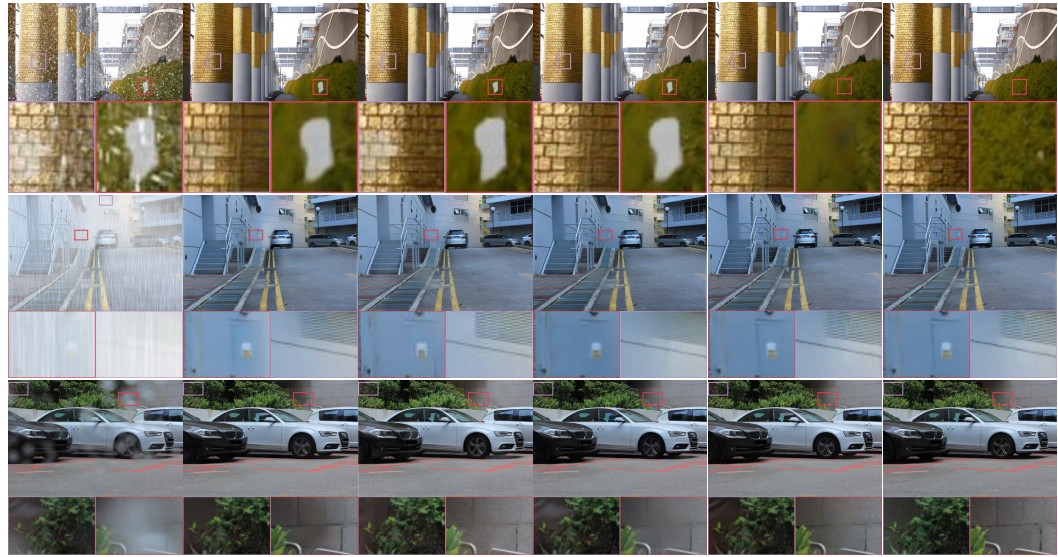

| (a) Input | (b) WGWSNet | (c) WeatherDiff | (d) Histoformer | (e) Ours | (f) GT |

Figure 8: Visual comparisons of MODEM against state-of-the-art unified methods across diverse adverse weather restoration tasks. **Top Row**: Image desnowing. **Middle Row**: Joint deraining and dehazing. **Bottom Row**: Raindrop removal.

Table 7: Ablation study results across different datasets and factor combinations.

| Factors | | | | Snow100K-S | | Snow100K-L | | Outdoor | | RainDrop | |
|---|---|---|---|---|---|---|---|---|---|---|---|
| Morton | DDEM | DAFM | DSAM | PSNR | SSIM | PSNR | SSIM | PSNR | SSIM | PSNR | SSIM |
| ✗ | ✓ | ✓ | ✓ | 38.03 | 0.9672 | 32.33 | 0.9283 | 32.89 | 0.9399 | 32.69 | 0.9423 |
| ✓ | ✗ | N/A | N/A | 37.61 | 0.9650 | 32.12 | 0.9249 | 32.37 | 0.9224 | 32.38 | 0.9390 |
| ✓ | ✓ | ✓ | ✗ | 38.15 | 0.9675 | 32.59 | 0.9298 | 32.77 | 0.9404 | 32.72 | 0.9435 |
| ✓ | ✓ | ✗ | ✓ | 37.43 | 0.9643 | 32.07 | 0.9240 | 32.19 | 0.9308 | 32.62 | 0.9387 |
| ✓ | ✓ | ✓ | ✓ | 38.08 | 0.9673 | 32.52 | 0.9292 | 33.10 | 0.9410 | 33.01 | 0.9434 |

yields the best results. This result is further supported by t-SNE [68] in Fig. 7. Compared to Histoformer [65], MODEM exhibits significantly improved clustering of features corresponding to different weather types. This indicates that the combined action of the components enables MODEM to learn more discriminative and well-separated feature representations, which directly contributes to enhance restoration capabilities. More ablation studies can be found in Appendix A.6.

# 6 Conclusion

In this paper, we introduced the Morton-Order Degradation Estimation Mechanism (MODEM) to address the challenge of restoring images degraded by diverse and spatially heterogeneous adverse weather. MODEM integrates a Dual Degradation Estimation Module (DDEM) for extracting global and local degradation priors, with a Morton-Order 2D-Selective-Scan Module (MOS2D) that employs Morton-coded spatial ordering and selective state-space models for adaptive, context-aware restoration. Extensive experiments demonstrate MODEM's state-of-the-art performance across multiple benchmarks and weather types. This superiority is attributed to its effective modeling of complex degradation dynamics via explicit degradation estimation guiding the restoration process, leading to more discriminative feature representations and strong generalization to real-world scenarios.

## Acknowledgement

We would like to thank Mingjia Li for the insightful discussions and feedback. This work was partially supported by the National Natural Science Foundation of China under Grant no. 62372251. The computational resources of this work was partially supported by TPU Research Cloud (TRC).

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

# A Appendix

## A.1 Implement Details

MODEM is implemented in PyTorch [56] and trained on 4 NVIDIA RTX 3090 GPUs. The main backbone contains residual groups with [4, 4, 6, 8, 6, 4, 4] MDSLs sequentially, and a final refinement group with 4 MDSLs. Downsampling and upsampling are performed using PixelUnshuffle and PixelShuffle, respectively. The initial channel size is 36. The DDEM employs residual groups with 2 MDSLs each, operating with 96 channels. We train MODEM in two stages. In Stage 1, DDEM takes the channel-concatenated degraded image $I_{LQ}$ and ground-truth $I_{GT}$ as input. We use the AdamW optimizer [47] with a learning rate of $3 \times 10^{-4}$, a weight decay of $1 \times 10^{-4}$, and betas set to $(0.9, 0.999)$. The learning rate is scheduled by a Cosine Annealing Restart Cyclic LR [46] scheduler with periods of [92k, 208k] iterations and restart weights of [1, 1]. The minimum learning rates, $\eta_{min}$, for these cycles are $[3 \times 10^{-4}, 1 \times 10^{-6}]$. Progressive training is adopted with patch sizes [128, 160, 256, 320, 384] and corresponding per-GPU mini-batch sizes [6, 4, 2, 1, 1] for [92k, 84k, 56k, 36k, 32k] iterations each, totaling 300k iterations. In Stage 2, DDEM processes only $I_{LQ}$, initializing parameters from Stage 1. The same AdamW optimizer settings and Cosine Annealing Restart Cyclic LR scheduler parameters are employed. Progressive training uses patch sizes [128, 160, 256, 320, 376] with per-GPU mini-batch sizes [8, 5, 2, 1, 1].

## A.2 Datasets and Metrics

To ensure a fair and comprehensive evaluation, MODEM is trained and tested consistent with those utilized in prior works [40, 67, 55, 65, 95, 79]. Our composite training data is aggregated from multiple sources, including 9,000 synthetic images featuring snow degradation from Snow100K [45], 1,069 real-world images affected by adherent raindrops from the Raindrop [57], and an additional 9,000 synthetic images from Outdoor-Rain [39] which are degraded by a combination of both fog and rain streaks. For performance evaluation, we utilize several distinct test sets: the Test1 [39, 40], the designated test split from the RainDrop [57], both the Snow100K-L and Snow100K-S subsets [45], and a challenging real-world test set from Snow100K comprising 1,329 images captured under various adverse weather conditions. We report PSNR and SSIM on these test datasets.

## A.3 Complexity Analysis

We report the parameters and inference time. The inference time is performed on a single Nvidia RTX 3090, with single $256 \times 256$ input image, detailed in Table 8.

Table 8: Comparison of parameters and inference time, along with average PSNR.

| Methods | WGWSNet [95] | WeatherDiff [55] | Histoformer [65] | MODEM (Ours) |
|---|---|---|---|---|
| Time (ms) | 24.83 | $1.67 \times 10^6$ | 109.07 | 92.86 |
| Parameters (M) | 2.65 | 82.96 | 16.62 | 19.96 |
| Average PSNR | 31.54 | 31.57 | 33.68 | 34.18 |

Table 9: Comparison of inference time (ms) for different input sizes and average PSNR.

| Input Size | WGWSNet [95] | WeatherDiff [55] | Histoformer [65] | MODEM (Ours) |
|---|---|---|---|---|
| $256 \times 256$ | 24.83 | $1.67 \times 10^6$ | 109.07 | 92.86 |
| $512 \times 512$ | 110.34 | $5.37 \times 10^6$ | 576.15 | 443.02 |
| $1024 \times 1024$ | 439.13 | $1.35 \times 10^7$ | 3056.29 | 1946.34 |
| Average PSNR | 31.54 | 31.57 | 33.68 | 34.18 |

To be more comprehensive, we evaluate the inference speed on larger resolutions in Table 9. As can be seen, compared to Transformer-based architectures like Histoformer [65], as the number of pixels increases, MODEM's complexity scales in a near-linear fashion, demonstrating significantly better scalability. This stands in contrast to the quadratic complexity often associated with Transformers, granting MODEM a distinct computational advantage.

Table 10: Comparison of parameters and inference time for other Mamba-style methods.

| Method | MambaIR [20] | FreqMamba [91] | MODEM (Ours) |
|---|---|---|---|
| Parameters (M) | 15.78 | 8.93 | 19.96 |
| Time (ms) | 790.61 | 233.41 | 92.86 |

Table 11: Comparison of inference time (ms) between Hilbert scan and Morton scan.

| Input Size | $256 \times 256$ | $512 \times 512$ | $1024 \times 1024$ |
|---|---|---|---|
| Hilbert scan [28] | 604.00 | 10134.91 | 88288.46 |
| Morton scan (Ours) | 92.86 | 443.02 | 1946.34 |

To further contextualize our model's performance, we compared MODEM with other Mamba-style methods, MambaIR [20] and FreqMamba [91]. As shown in Table 10, although MODEM has a larger parameter count compared to both MambaIR [20] and FreqMamba [91], it achieves a significantly faster inference time, highlighting the superior efficiency of our architectural design.

For scanning methods, Hilbert scan [28] is another locality-preserving alternative. However, it comes with a significantly higher computational cost, whereas the Morton-order can be calculated efficiently with simple bitwise operations as shown in our Eq. (6). To be clear, we compared the inference speed of the Morton and Hilbert scans. For the Hilbert scan, we used the official implementation from LC-Mamba [28]. The speed (ms) of different resolutions are shown in the Table 11, with all tests performed on a single RTX 3090 GPU. Our method is significantly faster than Hilbert.

## A.4 More Visualizations of Features

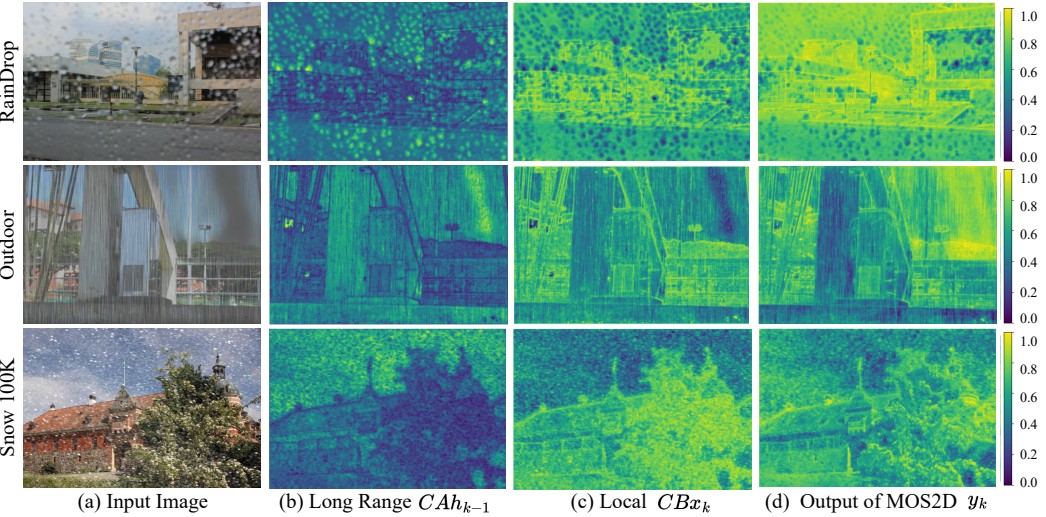

(a) Input Image     (b) Long Range $CAh_{k-1}$     (c) Local $CBx_k$     (d) Output of MOS2D $y_k$

Figure 9: With respect to a sample (a), (b)-(d) visualize the long-range $CAh_{k-1}$, (c) local $CBx_k$, and (d) output of MOS2D $y_k$, respectively.

## A.5 Limitations

While our proposed MODEM demonstrates state-of-the-art performance across a variety of adverse weather conditions, we acknowledge certain limitations. As illustrated in Fig. 10 of challenging real-world snow scenarios, MODEM, like other contemporary methods, can encounter difficulties in achieving perfect restoration when faced with images containing extremely large, high-contrast snowflakes. Such scenarios are particularly challenging if these specific visual patterns of snow, differing significantly in scale, density, or opacity from typical training examples, are underrepresented or entirely absent in the training data distribution. Nevertheless, empowered by its robust degradation estimation capabilities, MODEM still achieves comparatively better results in these extreme cases, effectively suppressing artifacts and preserving some structural detail. This highlights an ongoing

challenge in achieving perfect generalization to all unseen severe degradations, but also underscores the significant benefit of incorporating explicit and adaptive degradation estimation.

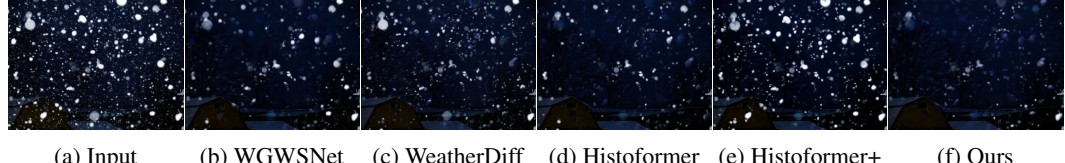

| (a) Input | (b) WGWSNet | (c) WeatherDiff | (d) Histoformer | (e) Histoformer+ | (f) Ours |

Figure 10: Visual comparison on a challenging real-world snow case from the dataset in [45], illustrating performance under severe degradation unseen during training. We compare MODEM to the prior methods [95, 55, 65], where "+" denotes additional training of Histoformer.

## A.6 More Ablation Studies

The Morton-order scans an image by processing it in small, contiguous block-like regions, moving from one adjacent block to the next (refer to the scale-space theory). This ensures that pixels that are close in the 2D image stay close in the 1D sequence. To further investigate the impact of the scanning scheme itself, we offer an ablation study within our MODEM, comparing the Morton scan against others [20, 74, 63, 25, 28, 24]. As shown in Table 12, our Morton achieves the best performance.

Table 12: Ablation study of diffrent scanning scheme.

| Methods | Snow100K-L | | Snow100K-S | | Outdoor | | RainDrop | |
|---|---|---|---|---|---|---|---|---|
| | PSNR | SSIM | PSNR | SSIM | PSNR | SSIM | PSNR | SSIM |
| Raster (MambaIR [20]) | 32.33 | 0.9283 | 38.03 | 0.9672 | 32.89 | 0.9399 | 32.69 | 0.9423 |
| Continuous (Zigzag [74]) | 32.17 | 0.9266 | 37.74 | 0.9662 | 32.35 | 0.9385 | 32.59 | 0.9422 |
| OSS (VmambaIR [63]) | 32.14 | 0.9262 | 37.39 | 0.9651 | 32.61 | 0.9387 | 32.11 | 0.9413 |
| Local (LocalMamba [25]) | 32.23 | 0.9266 | 37.75 | 0.9661 | 32.60 | 0.9382 | 32.53 | 0.9418 |
| Hilbert (LC-Mamba [28]) | 32.46 | 0.9287 | 37.96 | 0.9671 | 32.99 | **0.9414** | 32.82 | 0.9433 |
| Morton (Ours) | **32.52** | **0.9292** | **38.08** | **0.9673** | **33.10** | 0.9410 | **33.01** | **0.9434** |

We conduct an ablation study on the contribution of the loss terms. We have also performed an ablation study on the placement (before/after the average pooling) of the KL divergence loss. The results are presented in the table below. They confirm that each component contributes positively.

Table 13: Ablation study of the loss function.

| Methods | Snow100K-L | | Snow100K-S | | Outdoor | | RainDrop | |
|---|---|---|---|---|---|---|---|---|
| | PSNR | SSIM | PSNR | SSIM | PSNR | SSIM | PSNR | SSIM |
| w/o Correlation Loss | 32.30 | 0.9278 | 37.79 | 0.9668 | 32.91 | 0.9403 | 32.69 | 0.9427 |
| w/o KL Loss | 32.12 | 0.9249 | 37.61 | 0.9650 | 32.37 | 0.9224 | 32.38 | 0.9390 |
| Replace KL Loss | 31.85 | 0.9237 | 37.16 | 0.9638 | 32.77 | 0.9389 | 32.57 | 0.9387 |
| MODEM | **32.52** | **0.9292** | **38.08** | **0.9673** | **33.10** | **0.9410** | **33.01** | **0.9434** |

## A.7 More Quantitative Comparisons

We retrain three prominent Mamba-style restoration networks [20, 63, 96] on the all-weather setting. These experiments were conducted carefully following the settings used by other recent methods like ConvIR [13], FSNet [92], and Histoformer [65], while also respecting their original training configurations. The results are presented in Table 14. Our MODEM achieves the best performance.

Table 14: Comparison with Mamba-like Methods.

| Methods | Snow100K-L | | Snow100K-S | | Outdoor | | RainDrop | |
|---|---|---|---|---|---|---|---|---|
| | PSNR | SSIM | PSNR | SSIM | PSNR | SSIM | PSNR | SSIM |
| MambaIR [20] | 28.59 | 0.8729 | 33.34 | 0.9330 | 24.73 | 0.8808 | 32.16 | 0.9393 |
| FreqMamba [96] | 27.09 | 0.8624 | 33.52 | 0.9331 | 19.89 | 0.7519 | 30.60 | 0.9198 |
| VmambaIR [63] | 31.07 | 0.9168 | 36.35 | 0.9605 | 24.23 | 0.8558 | 32.18 | 0.9392 |
| MODEM | **32.52** | **0.9292** | **38.08** | **0.9673** | **33.10** | **0.9410** | **33.01** | **0.9434** |

## A.8 More Visual Comparisons

Further visual comparisons are presented in Figs. 11 to 14.

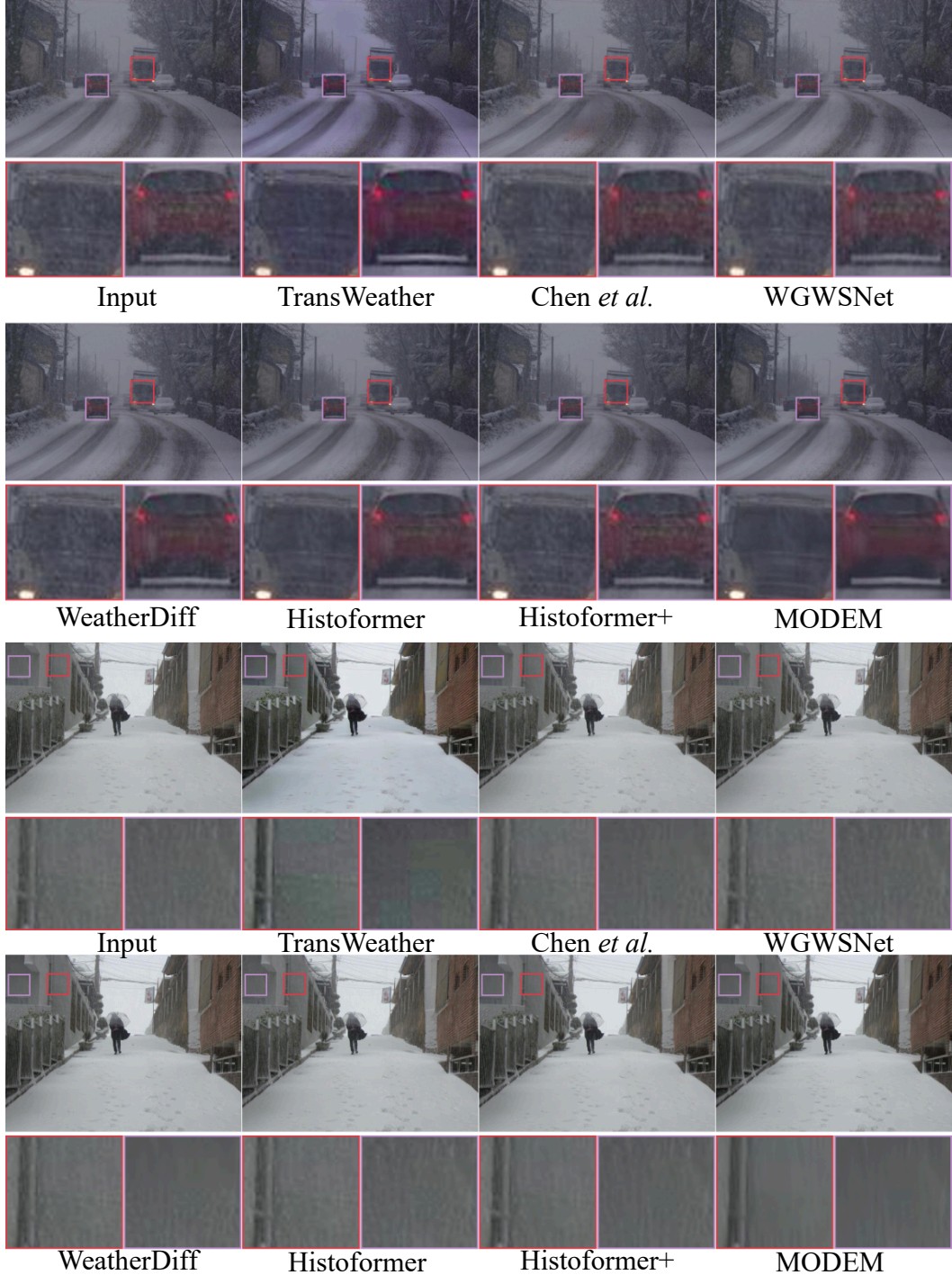

Figure 11: More visual results for desnowing on real-world snowy images [45].

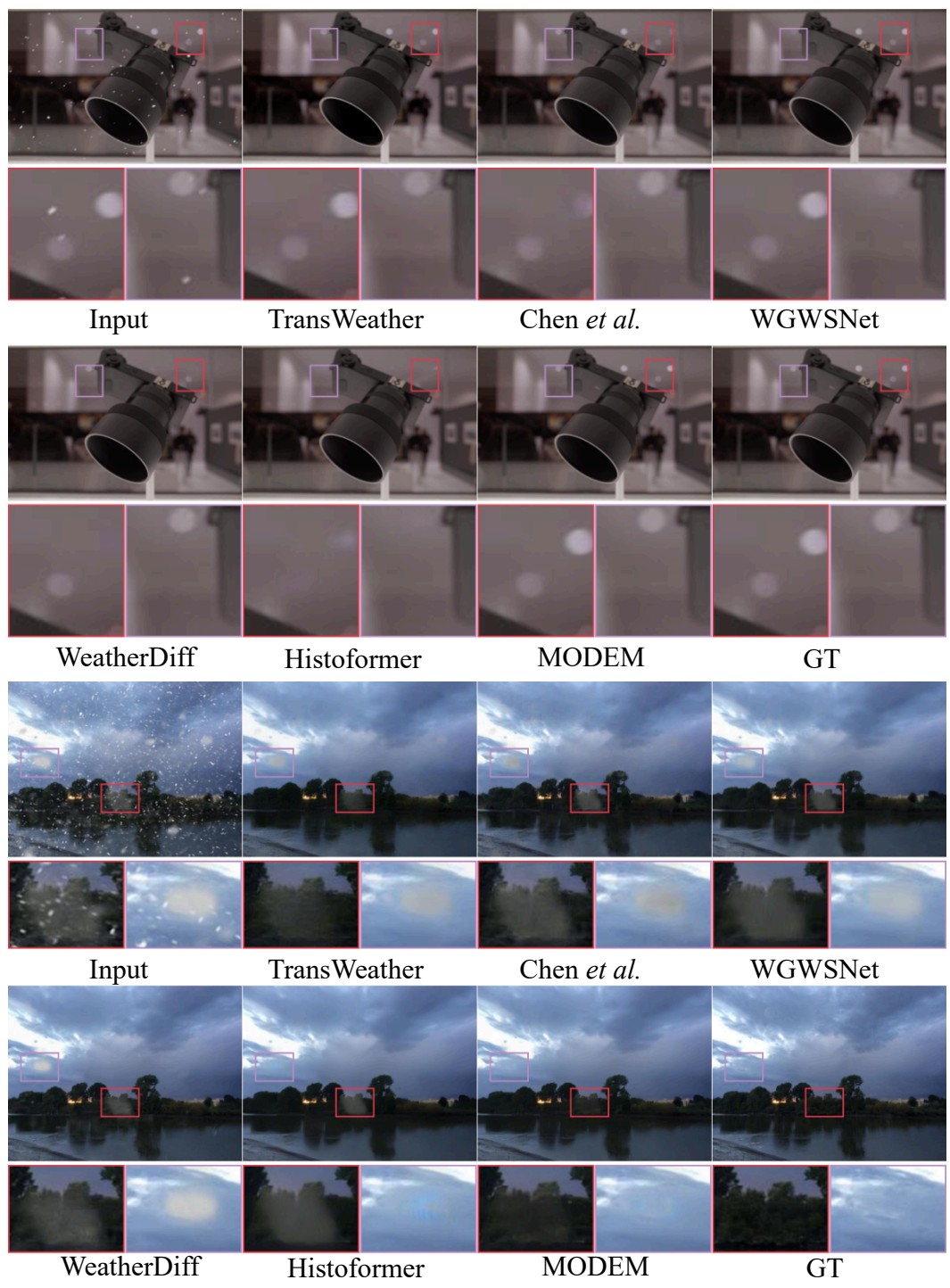

Figure 12: More visual results for image desnowing on the Snow100K [45].

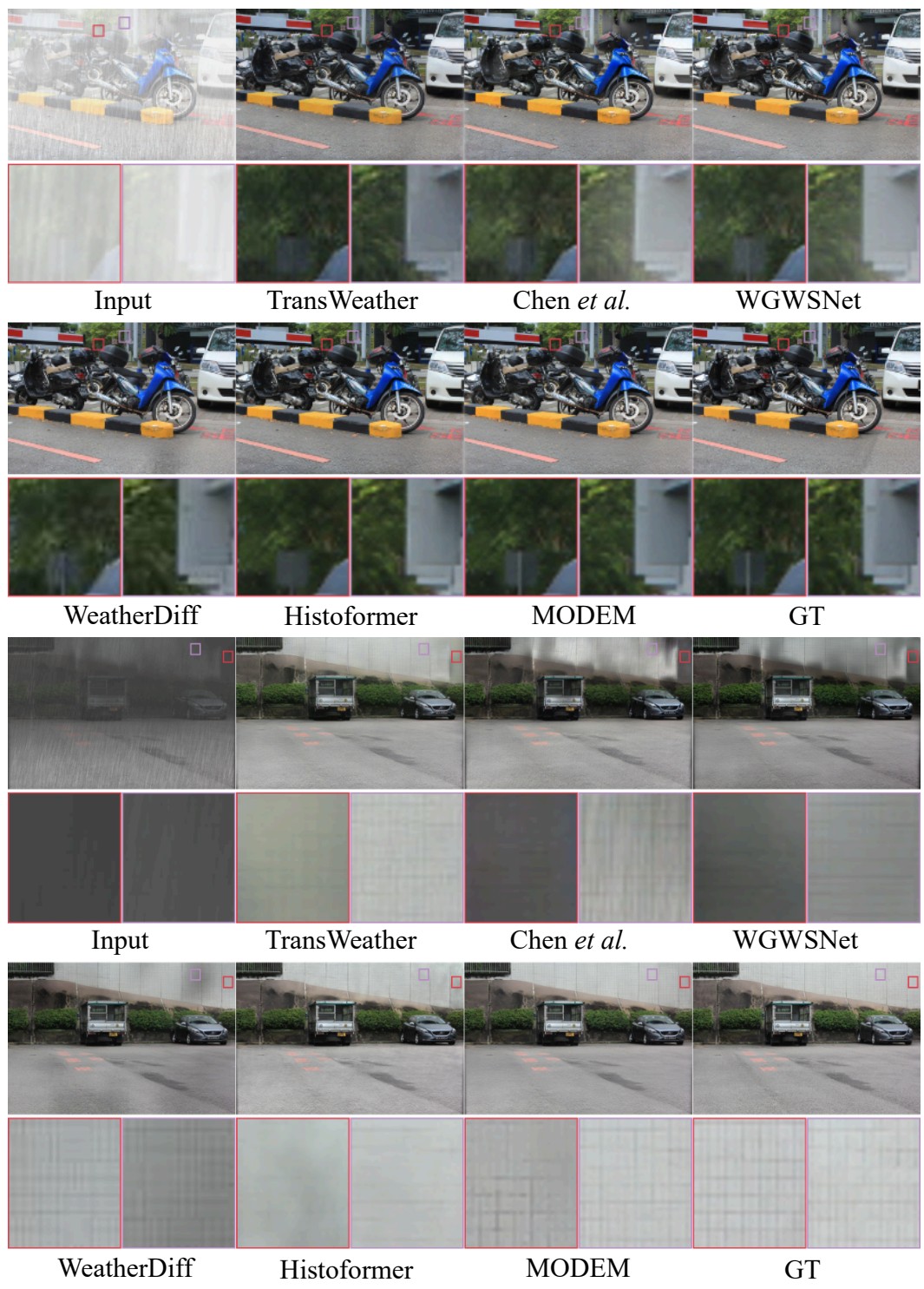

| Input | TransWeather | Chen *et al.* | WGWSNet |
| WeatherDiff | Histoformer | MODEM | GT |
| Input | TransWeather | Chen *et al.* | WGWSNet |
| WeatherDiff | Histoformer | MODEM | GT |

Figure 13: More visual results for deraining&dehazing on the Test1 dataset [39, 40].

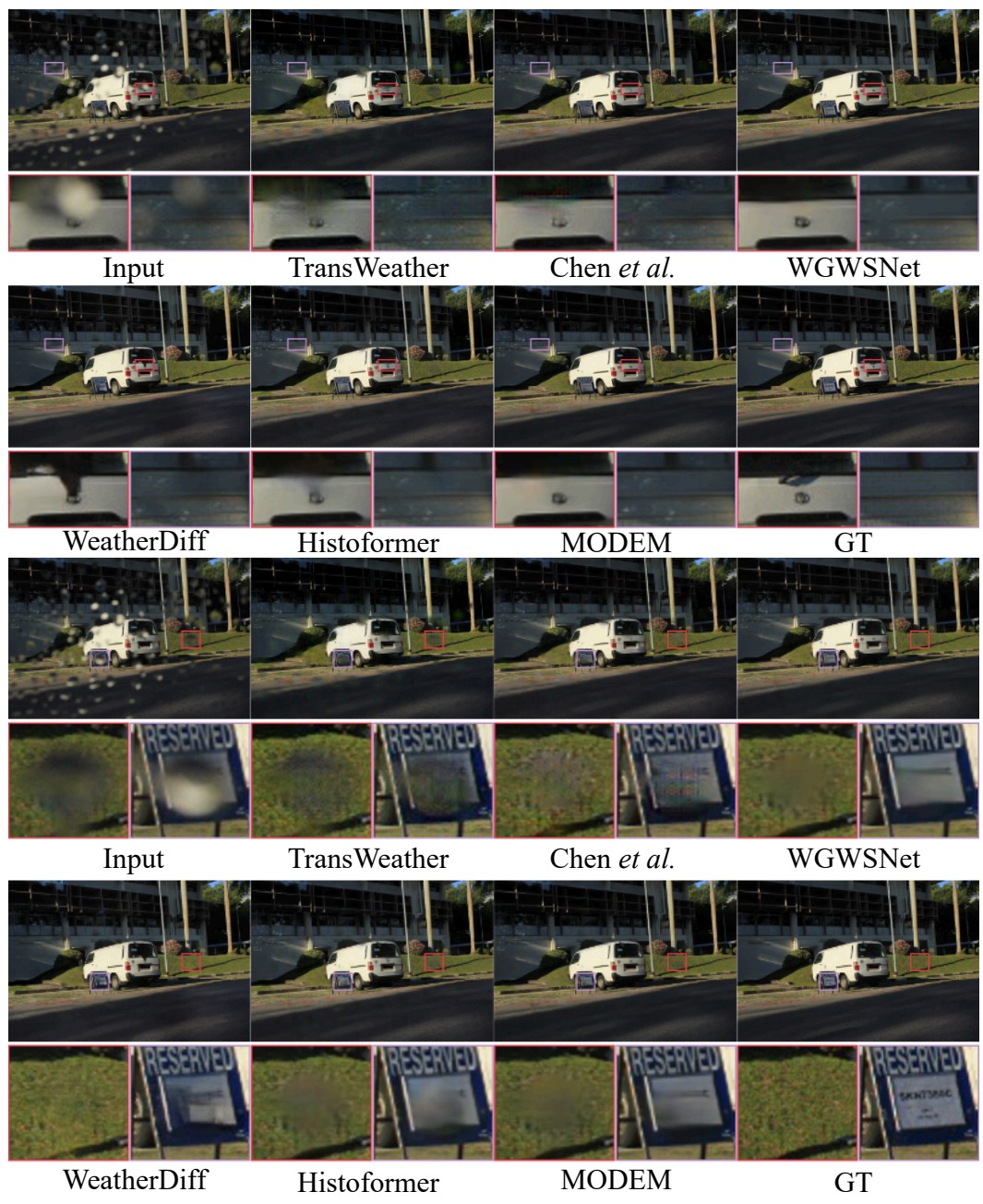

Figure 14: More visual results for raindrop removal on the Raindrop dataset [57].

