# OpenReview forum: "MODEM: A Morton-Order Degradation Estimation Mechanism for Adverse Weather Image Recovery"
_NeurIPS.cc/2025/Conference — NeurIPS 2025 poster_

### Official Review · Reviewer_2iWh · 2025-06-21

**Clarity:** 3
**Significance:** 2
**Originality:** 3
**Rating:** 4
**Confidence:** 4

**Summary:**

This paper introduces a Morton-Order Degradation Estimation Mechanism for adverse weather image restoration. A Morton-Order 2D-Selective-Scan Module is developed to capture long-range dependencies while preserving local structural coherence. Meanwhile, a Dual Degradation Estimation Module is proposed to disentangles degradations. The idea of employing SSM is recently popular in this field, and the newly designed model sounds interesting.

**Questions:**

1) The authors are encouraged to include comparisons with more recent models in Tables 2–4, especially works from 2023–2024 such as ConvIR[3], AST[4], CODE[5], and FSNet[6], which are highly relevant in this context.

2) In Tab.1, a comparison with the diffusion-based method in terms of PSNR/SSIM is not enough, more perceptual metric is supposed to be provided, such as LPIPS.

3) Complexity Analysis is conducted, while no SSM-based methods are included, such as MambaIR[1], FreqMamba[2].

4) Effective Receptive Field (ERF) visualization of MODEM and competitors (e.g., MambaIR[1]) would help in understanding their long-range modeling ability.

5) Quantitative results (such as non-reference metric, NIQE) or a user study on challenging real-world scenes should be conducted.

6) The ablation study could be extended to explore the contribution of different loss components

[3] Revitalizing Convolutional Network for Image Restoration. TPAMI24.

[4] Adapt or perish: Adaptive sparse transformer with attentive feature refinement for image restoration. CVPR24.

[5] Comprehensive and delicate: An efficient transformer for image restoration. CVPR23.

[6] Image Restoration via Frequency Selection. TPAMI24.

**Ethical Concerns:**

["NO or VERY MINOR ethics concerns only"]

**Final Justification:**

After reading the comments from other reviewers and the replies made by authors, I would like to maintain my score. While this paper needs more details to help understand, which is expected to be addressed in the coming version, the idea of this paper itself is appreciated.

**Limitations:**

Limitation of the method is not discussed.

**Quality:**

2

**Strengths And Weaknesses:**

Pros:

1) Using Morton Scanning instead of standard Scanning ways brings new insight to this community.

2) The proposed MODEM performs favorably against the considered methods.

Cons:

1) Differences between MODEM and existing SSM-based methods (e.g., MambaIR[1], FreqMamba[2]) are supposed to be further analyzed.

2) The space/time complexity of the proposed algorithm is not studied.

[1] Mambair: A simple baseline for image restoration with state-space model. ECCV24.

[2] Freqmamba: Viewing mamba from a frequency perspective for image deraining. ACM MM24.

---

> ### Author Rebuttal · Authors · 2025-07-31
>
> **Q1: Differences between** **MODEM** **and existing SSM-based methods? (Cons1)**
>
> **A1:** Thank you for this suggestion. The key difference is our **explicit degradation estimation**. While methods like MambaIR and FreqMamba are "blind" backbones, MODEM features a **Dual Degradation Estimation Module (DDEM)**. The DDEM extracts a **global degradation descriptor** and **spatially adaptive kernels (Eq. 4)**. These degradation priors then dynamically modulate our MOS2D modules via the proposed **DAFM (Eq. 7)** and **DSAM (Eq. 8)** mechanisms, which in turn enables a degradation-aware selective scan process by dynamically modulating the SSM **(Eq. 9, Eq. 10)**.
>
> To address this concern, we have retrained three prominent Mamba-style restoration networks on the all-weather setting. These experiments were conducted carefully following the settings used by other recent methods like ConvIR, FSNet, and Histoformer, while also respecting their original training configurations. The results are presented in the table below.
>
> | Method    | Snow100K-L | Snow100K-S | Outdoor   | Raindrop  |
> | --------- | ---------- | ---------- | --------- | --------- |
> | MambaIR   | 28.59      | 33.34      | *24.73*   | 32.16     |
> | FreqMamba | 27.09      | 33.52      | 19.89     | 30.60     |
> | VmambaIR  | *31.07*    | *36.35*    | 24.23     | *32.18*   |
> | MODEM     | **32.52**  | **38.08**  | **33.10** | **33.01** |
>
> Furthermore, Unlike the simple raster scan used in MambaIR, our MOS2D uses the **locality-preserving Morton scan**. This is specifically designed to enable more structured and effective modeling of the spatially heterogeneous nature of weather degradation. Our ablation in Table 5 already showed its advantage over a raster scan. To further validate this, we conducted an expanded ablation study comparing Morton-order against other scanning patterns.
>
> | Method               | Snow100K-L | Snow100K-S | Outdoor   | Raindrop  |
> | -------------------- | ---------- | ---------- | --------- | --------- |
> | Raster (e.g.MambaIR) | 32.33      | 38.03      | 32.89     | 32.69     |
> | Continuous (zigzag)  | 32.17      | 37.74      | 32.35     | 32.59     |
> | OSS (VmambaIR)       | 32.14      | 37.39      | 32.61     | 32.11     |
> | Local (LocalMamba)   | 32.23      | 37.75      | 32.60     | 32.53     |
> | Hilbert (LC-Mamba)   | *32.46*    | *37.96*    | *32.99*   | *32.82*   |
> | Morton (Ours)        | **32.52**  | **38.08**  | **33.10** | **33.01** |
>
> We will include these ablation and comparison experiments in the revision.
>
> **Q2: Complexity Analysis? (Cons2 & Q3)**
>
> **A2:** Thank you for this suggestion. In Appendix A.3, we initially provided a complexity analysis comparing our model to the previous SOTA methods in the unified all-weather setting. We did not include MambaIR and FreqMamba as their papers use different benchmarks than our all-weather evaluation.
>
> To address your concern, we have now conducted a direct complexity comparison with MambaIR and FreqMamba under identical settings (256x256 input, single RTX 3090).
>
> | Method         | MambaIR | FreqMamba | Histoformer | MODEM |
> | -------------- | ------- | --------- | ----------- | ----- |
> | Parameters (M) | 15.78   | 8.93      | 16.62       | 19.96 |
> | Time (ms)      | 790.61  | 233.41    | 109.07      | 92.86 |
>
> The results show MODEM has a superior balance of performance and efficiency, achieving significantly faster inference than other SSM-based methods. This will be added to the revision.
>
> **Q3: More comparisons? (Q1)**
>
> **A3:** Thank you for suggesting these highly relevant and recent models for comparison. Due to the time constraints of the rebuttal period, we have focused on conducting new experiments on the tasks where our benchmarks overlap with these newer methods. It is important to note that in the all-weather setting, the "RainDrop" benchmark is for raindrop removal, and the "Outdoor" benchmark is for joint rain and haze removal. Since ConvIR and FSNet do not address the tasks of our RainDrop or Outdoor benchmarks, we have compared our method against them on the Snow100K dataset. To ensure a rigorous comparison, we evaluated on both the Snow100K-L/S from the all-weather setting and the Snow100K-2000 used in the ConvIR and FSNet papers. The comparative results are presented in the table below.
>
> | Method | Snow100K-L | Snow100K-S | Snow100K-2000 |
> | ------ | ---------- | ---------- | ------------- |
> | ConvIR | *32.11*    | *37.98*    | *33.92*       |
> | FSNet  | 31.62      | 37.42      | 33.76         |
> | MODEM  | **32.52**  | **38.08**  | **35.76**     |
>
> For AST, there is an overlap in the RainDrop removal task. We have therefore tested AST on our RainDrop benchmark, and the results are shown in the table below.
>
> | Method | AST   | MODEM     |
> | ------ | ----- | --------- |
> | PSNR   | 30.57 | **33.01** |
>
> There were no directly overlapping tasks with CODE. Therefore, we have retrained CODE separately on each of the three tasks within the all-weather dataset, carefully following the settings used by other recent methods like ConvIR, FSNet, and Histoformer, while respecting CODE's original training configuration. The results of this comparison are shown in the table below.
>
> | Method | Snow100K-L | Snow100K-S | Snow100K-2000 | Outdoor   | Raindrop  |
> | ------ | ---------- | ---------- | ------------- | --------- | --------- |
> | CODE   | 30.46      | 36.29      | 33.90         | 26.65     | 32.64     |
> | MODEM  | **32.52**  | **38.08**  | **35.76**     | **33.10** | **33.01** |
>
> We have made our best effort to present these new and relevant comparisons given the limited time. We commit to conducting a more exhaustive comparison for the revision.
>
> **Q4: More perceptual metric? (Q2 & Q5)**
>
> **A4:** Thank you for this insightful suggestion. We have added new perceptual metrics. We now report referenced (LPIPS) and non-referenced (Q-Align, MUSIQ) scores, all computed using the `pyiqa` library, with the same settings applied for all methods. As shown in the table below, our method achieves SOTA perceptual quality.
>
> Q-Align, higher is better.
>
> | Method      | Snow100K-L | Snow100K-S | Outdoor    | Raindrop   | Snow100K-Real |
> | ----------- | ---------- | ---------- | ---------- | ---------- | ------------- |
> | MODEM       | **3.7324** | **3.7695** | **4.1875** | **4.0664** | **3.5586**    |
> | Histoformer | *3.7207*   | *3.7598*   | *4.1445*   | *4.0156*   | *3.5449*      |
> | WeatherDiff | 3.4531     | 3.5293     | 3.8691     | 4.0000     | 3.4512        |
>
> MUSIQ, higher is better.
>
> | Method      | Snow100K-L  | Snow100K-S  | Outdoor     | Raindrop    | Snow100K-Real |
> | ----------- | ----------- | ----------- | ----------- | ----------- | ------------- |
> | MODEM       | *64.2438*   | **64.2853** | **68.2926** | **69.7925** | **59.6042**   |
> | Histoformer | **64.2526** | *64.2581*   | *67.7461*   | 68.4852     | 59.4040       |
> | WeatherDiff | 62.6267     | 63.1278     | 67.4814     | *69.3608*   | *59.4493*     |
>
> LPIPS, lower is better.
>
> | Method      | Snow100K-L | Snow100K-S | Outdoor    | Raindrop   |
> | ----------- | ---------- | ---------- | ---------- | ---------- |
> | MODEM       | **0.0880** | **0.0407** | **0.0699** | *0.0650*   |
> | Histoformer | *0.0919*   | *0.0445*   | *0.0778*   | 0.0672     |
> | WeatherDiff | 0.0982     | 0.0541     | 0.0887     | **0.0615** |
>
> **Q5: Effective** **Receptive Field** **(ERF) visualization? (Q4)**
>
> **A5:** Thank you for this suggestion. Due to the rules of the rebuttal process, we are unable to upload new ERF visualizations at this stage. We have calculated the size of the ERF for our model and key competitors on a 256x256 input image. We measured the ERF at different contribution thresholds (1%, 5%, 10%, and 15%). The results are presented in the table below:
>
> | Threshold   | 1%        | 5%       | 10%      | 15%      |
> | ----------- | --------- | -------- | -------- | -------- |
> | MODEM       | 243 x 200 | 129 x 67 | 117 x 57 | 116 x 57 |
> | Histoformer | 168 x 250 | 33 x 26  | 7 x 6    | 2 x 1    |
> | MambaIR     | 171 x 232 | 50 x 125 | 13 x 21  | 10 x 18  |
>
> These results confirm MODEM's larger effective receptive field and superior long-range modeling compared to competitors. We will include the full ERF visualizations in the final camera-ready version of the paper.
>
> **Q6: Ablation study of different loss components? (Q6)**
>
> **A6:** Thank you for your suggestion. The importance of the KL loss was already shown in our Table 5 ablation (row 2). The results show the critical supervisory role of the KL divergence for the degradation estimation process. To further analyze our loss function, we now provide new ablations on the correlation loss and the KL loss placement (before the avg. pool), as shown in the table below:
>
> | Setup                | Snow100K-L | Snow100K-S | Outdoor   | Raindrop  |
> | -------------------- | ---------- | ---------- | --------- | --------- |
> | w/o Correlation Loss | 32.30      | 37.79      | 32.91     | 32.69     |
> | w/o KL Loss          | 32.12      | 37.61      | 32.37     | 32.38     |
> | Replace KL Loss      | 31.85      | 37.16      | 32.77     | 32.57     |
> | MODEM                | **32.52**  | **38.08**  | **33.10** | **33.01** |
>
> These results confirm that each loss component contributes positively to the final performance. We will incorporate this more detailed ablation study into the revision.
>
> **Q7: Limitations?**
>
> **A7:** We analyze our model's limitations, including failure cases, in Appendix A.4 (Figure 8). As we discuss there, MODEM, like other contemporary methods, can struggle with images containing extremely large, high-contrast snowflakes. Nevertheless, as the visual results show, our method still achieves comparatively better results in these challenging cases. We will reference this analysis in the main text.

---

> > ### Comment · Reviewer_2iWh · 2025-08-04
> > **Review of MODEM**
> >
> > Thanks for the rebuttal made by the authors. After reading the comments from other reviewers and the replies made by authors, I would like to maintain my score. While this paper needs more details to help understand, which is expected to be addressed in the coming version, the idea of this paper itself is appreciated.

---

> > > ### Author Response · Authors · 2025-08-04
> > > **Reply to Reviewer 2iWh**
> > >
> > > Thank you for your positive feedback on our idea and for providing these valuable suggestions for comparison. We will be sure to incorporate the experiments and details you have recommended into the revised version of our paper.

---

### Official Review · Reviewer_6wM5 · 2025-06-21

**Clarity:** 2
**Significance:** 2
**Originality:** 2
**Rating:** 3
**Confidence:** 4

**Summary:**

This paper proposes the Morton-Order Degradation Estimation Mechanism (MODEM) for adverse weather removal. MODEM integrates a Dual Degradation Estimation Module (DDEM) for extracting global and local degradation priors, with a Morton-Order 2D-Selective-Scan Module (MOS2D) that employs Morton-coded spatial ordering and selective state-space models for adaptive, context-aware restoration. Extensive experiments demonstrate MODEM’s state-of-the-art performance across multiple benchmarks and weather types.

**Questions:**

See the Weaknesses

**Ethical Concerns:**

["NO or VERY MINOR ethics concerns only"]

**Final Justification:**

Thanks for your rebuttal. I think that this paper does not have significant novelty for the following reasons:

1, Morton mechanism is not a novel idea. The authors attempt to combine the mamba and Morton.

2, For other modules, such as degradation estimation, are also not new. Various works have been explored. The novelty is incremental.

3, The compared methods in Tables 2,3,4 are out of date, which is not acceptable, although the authors add new experiments during the rebuttal.

I thus maintain my score.

**Limitations:**

yes.

**Paper Formatting Concerns:**

None.

**Quality:**

3

**Strengths And Weaknesses:**

Strengths.

1, This paper is well-written and easy to follow. Moreover, this paper has clear structure and is easy to understand.

2, Morton-Order Degradation Estimation Mechanism seems to be interesting, which integrates Morton-coded spatial ordering with selective state space modeling to effectively capture spatially heterogeneous weather degradation dynamics.

Weaknesses.
I find that this paper exists some major concerns.

1, Figure 2 and Figure 2 may be inappropriate. First of all, in Figure 2, the clear image is not equal to the sum of these three component. Putting it in the paper may confuse readers. Moreover, I do not find the significant meaning for putting the Figure 3 in current position. It may be meaningful but not major. The authors should consider to replace these two figures with more convincing demonstration.

2, I also find that in Tables 2-4,  the authors compare with latest method which is published in 2022. The authors should compare with recent state-of-the-art approaches. Comparing with outdate methods is not acceptable.

3, I observe that, in Table 5, the full model is not better than the model without DSAM in the datasets of Snow100K-S and Snow100K-L. The authors should give an in-depth analysis and explanations.

4, The authors do not give the analysis about the two-stage training mechanism. How about only training the first stage without using GT within the same training epoch? Where does the correlation loss derive from? Why not the authors analyze this? Why the authors computing the KL loss at the position of $\tilde{Z}$? Why not compute the vectors at the position of the feature of Figure 4(b), i.e., the feature before the operation of avg pool.

5, The authors also should compare the results of different scanning manners including, (a) Raster Scanning (b) Continuous Scanning (c) Local Scanning, which is illustrated in Figure 1.

---

> ### Author Rebuttal · Authors · 2025-07-31
>
> **Q1: Figure 2 and Figure 3? (W1)**
>
> **A1:** Thank you for this insightful feedback. The intent of Figure 2 is to provide an intuitive, visual analogy for the terms in Eq. 3 when considering the residual term $Dx_k$. The output $y_k$ in the diagram is meant to represent the output of the S6 block at step $k$. **We used a clean image to visually suggest that this output is a feature representation closely related to the restored clean image.** Due to the constraints of the rebuttal process, we are unable to upload a revised figure at this time, but we will provide a more rigorous illustration in the revision of the paper.
>
> The Figure 3 provided empirical evidence of our model's behavior by visualizing the intermediate feature maps. Specifically, it shows the feature distributions for the key terms within the MOS2D's S6 block. As analyzed in Section 3, we hypothesized that the $CAh_{k-1}$ term would focus more on long-range degradation information (in the case of Figure 3, the raindrops), while the $CBx_k$ term would focus more on local information, including local image content like fine textures. The visualizations in Figure 3 confirm that our trained model indeed learns the information in this manner. We will add examples from the other tasks in the revision for a more comprehensive demonstration.
>
> **Q2: More comparisons? (W2)**
>
> **A2:** Thank you for this suggestion. We would like to clarify that the all-weather setting is primarily designed for unified models. Consequently, many excellent recent restoration methods have not been evaluated in these benchmarks, which is why they were not originally included.
>
> To address your concern, we have now benchmarked our method against the recent SOTA approaches, including ConvIR, AST, CODE, FSNet, and the Mamba-based models like MambaIR, VMambaIR, and FreqMamba.
>
> Since ConvIR and FSNet do not address the RainDrop (raindrop removal) or Outdoor (joint rain-haze) benchmarks, we compared them on the overlapping desnowing task. To ensure a rigorous comparison, we evaluated on both the Snow100K-L/S benchmarks from the all-weather setting and the Snow100K-2000 benchmark from the ConvIR and FSNet, as shown below.
>
> | Method | Snow100K-L | Snow100K-S | Snow100K-2000 |
> | ------ | ---------- | ---------- | ------------- |
> | ConvIR | 32.11      | 37.98      | 33.92         |
> | FSNet  | 31.62      | 37.42      | 33.76         |
> | MODEM  | **32.52**  | **38.08**  | **35.76**     |
>
> For AST, there is an overlap in the RainDrop removal task. We have therefore tested AST on the RainDrop benchmark, and the results are shown in the table below.
>
> | Method | RainDrop  |
> | ------ | --------- |
> | AST    | 30.57     |
> | MODEM  | **33.01** |
>
> For methods without task overlap (CODE, MambaIR, VMambaIR, FreqMamba), we retrained them on each of the tasks, carefully following the settings used by other recent methods like ConvIR, FSNet, and Histoformer, while respecting their original training configurations. The results are shown in the table below.
>
> | Method    | Snow100K-L | Snow100K-S | Outdoor   | Raindrop  |
> | --------- | ---------- | ---------- | --------- | --------- |
> | MambaIR   | 28.59      | 33.34      | 24.73     | 32.16     |
> | FreqMamba | 27.09      | 33.52      | 19.89     | 30.60     |
> | VmambaIR  | 31.07      | 36.35      | 24.23     | 32.18     |
> | CODE      | 30.46      | 36.29      | 26.65     | 32.64     |
> | MODEM     | **32.52**  | **38.08**  | **33.10** | **33.01** |
>
> We will fully integrate these more comprehensive comparisons into Tables 2-4 in the revision of the paper.
>
> **Q3: Analysis of ablation study? (W3)**
>
> **A3:** Thank you for prompting a deeper discussion on this interesting result. Firstly, the all-weather training data is imbalanced, with fewer images for raindrop removal and deraining/dehazing than for desnowing. The DSAM empowers the model to handle the **fine-grained, localized artifacts** characteristic of these under-represented tasks. Without DSAM's guidance, the network would naturally over-specialize on the more prevalent snow degradation.
>
> As we analyzed in Section 5.2 for Table 5, this trade-off is evident in the results. Removing DSAM (row 3) results in a marginal performance increase on the snow datasets (+0.07dB on both Snow100K-S/L). However, this comes at a significant cost to the other tasks: performance drops by 0.33dB on the Outdoor dataset and by 0.29dB on the RainDrop dataset. This clearly shows that **DSAM provides the crucial modulation capability for the model to adapt to fine-grained, local artifacts**, preventing it from over-specializing on more globally uniform degradations. We will clarify this point in the revision.
>
> **Q4: Two-stage training mechanism? (W4)**
>
> **A4:** Thank you for these insightful questions. The primary purpose of our two-stage training is to effectively solve the difficult problem of degradation estimation. **In Stage 1,** we train the DDEM to learn an ideal degradation representation by leveraging the paired GT-degraded images from the training set. Specifically, the DDEM module receives both images as input and is trained to understand their difference, ultimately outputting a single, compact degradation vector. This vector establishes the **perfect degradation prior** that will serve as the supervisory target for Stage 2.
>
> **In Stage 2,** we use this learned degradation prior to supervise the training of our DDEM module via KL divergence, as shown in Eq. 13. Specifically, a **frozen copy of the Stage 1 DDEM** is used to generate the target degradation priors from the paired training images. The practical DDEM, which receives only the degraded image as input, is then trained to match the output of this frozen module. This two-stage training process ensures the MODEM can robustly estimate degradation.
>
> As shown in our ablation (Table 5, row 2), removing the degradation estimation link severely degrades performance, proving the superiority of our two-stage approach.
>
> **Q5: Compute the** **KL** **loss before the avg pool? (W4)**
>
> **A5:** As we analyzed in our introduction, learning a discriminative degradation representation directly from feature maps is notoriously difficult **because scene content and degradation are deeply entangled**. This is precisely why methods without explicit degradation estimation often struggle on multi-degradation tasks.
>
> Our solution, as described in the introduction, is to disentangle this problem by learning: (i) a global degradation representation that encapsulates high-level weather characteristics, and (ii) a set of spatially adaptive kernels that encode local degradation structures and variations. The effectiveness of these components is demonstrated in the 2nd-5th rows of our ablation study in Table 5.
>
> The vector $\tilde{Z}$ is a compact, global descriptor designed to capture the high-level degradation distribution of the entire image. By computing the KL divergence loss on $\tilde{Z}$, the model is guided to learn a disentangled, conceptual summary of the degradation, avoiding the challenges of a direct feature-map-level replication where content and degradation are deeply mixed. This encourages better generalization by focusing the learning on high-level degradation characteristics, which is validated by our model's strong performance on real-world images (Fig. 6) without any fine-tuning.
>
> Additionally, to further address your concern, we have conducted an experiment applying the KL loss at the feature-map level before average pooling, shown in the table below.
>
> | Setup           | Snow100K-L | Snow100K-S | Outdoor   | Raindrop  |
> | --------------- | ---------- | ---------- | --------- | --------- |
> | w/o KL Loss     | 32.12      | 37.61      | 32.37     | 32.38     |
> | Replace KL Loss | 31.85      | 37.16      | 32.77     | 32.57     |
> | MODEM           | **32.52**  | **38.08**  | **33.10** | **33.01** |
>
> **Q6: Correlation loss? (W4)**
>
> **A6:** We adopted correlation loss following its implementation in Histoformer. Its purpose is to better preserve patch-level linear correlations. We will add the citation to the revision of the paper.
>
> **Q7: Compare the different scanning manners? (W5)**
>
> **A7:** Thank you for this suggestion. We had already provided an initial comparison in our ablation study. Our ablation study **(Table 5, row 1)** shows that replacing our Morton scan with a Raster scan (used in MambaIR) degrades performance.
>
> To more comprehensively address your concern, we have now conducted an expanded ablation study to include all the scanning methods illustrated in Figure 1, as well as other relevant methods from recent works. We maintained the same MODEM architecture and only replaced the scanning mechanism, with each variant trained under the same settings. The results are presented in the table below:
>
> | Method               | Snow100K-L | Snow100K-S | Outdoor   | Raindrop  |
> | -------------------- | ---------- | ---------- | --------- | --------- |
> | Raster (e.g.MambaIR) | 32.33      | 38.03      | 32.89     | 32.69     |
> | Continuous (zigzag)  | 32.17      | 37.74      | 32.35     | 32.59     |
> | OSS (VmambaIR)       | 32.14      | 37.39      | 32.61     | 32.11     |
> | Local (LocalMamba)   | 32.23      | 37.75      | 32.60     | 32.53     |
> | Hilbert (LC-Mamba)   | 32.46      | 37.96      | 32.99     | 32.82     |
> | Morton (Ours)        | **32.52**  | **38.08**  | **33.10** | **33.01** |
>
> The results confirm that the Morton-order scan consistently performs best, providing strong evidence that its superior 2D locality preservation. We will incorporate this detailed ablation study into the revision.

---

> > ### Comment · Reviewer_6wM5 · 2025-08-05
> >
> > Thanks for your rebuttal. I think that this paper does not have significant novelty for the following reasons:
> >
> > 1, Morton mechanism is not a novel idea. The authors attempt to combine the mamba and Morton.
> >
> > 2, For other modules, such as degradation estimation, are also not new. Various works have been explored. The novelty is incremental.
> >
> > 3, The compared methods in Tables 2,3,4 are out of date, which is not acceptable, although the authors add new experiments during the rebuttal.
> >
> > I thus maintain my score.

---

> > > ### Author Response · Authors · 2025-08-05
> > > **Reply to Reviewer 6wM5**
> > >
> > > Thank you for your feedback and for your effort in reviewing our submission and rebuttal. We would like to offer the following clarifications in response to the concerns you raised:
> > >
> > > **On Novelty:**
> > >
> > > We appreciate your initial comment that our degradation estimation mechanism "seems to be interesting." Our primary contribution is the novel **combination of a Morton-ordered Mamba with a dedicated local and global degradation estimation framework.**
> > >
> > > While the individual components may have been explored in different contexts, their effective integration for this challenging task is non-trivial. The core novelty lies in demonstrating **how** the superior locality preservation of the Morton scan can be leveraged by a state-space model that is dynamically guided by learned degradation priors. Specifically, our **DDEM** learns to extract these global and local priors, which are then used to modulate the feature transformations within our **MOS2D** module. This synergy creates a degradation-aware selective scan process, allowing the model to effectively disentangle and adapt to the spatially heterogeneous artifacts present in diverse weather conditions. We respectfully point out that many impactful contributions in vision are built upon known components, e.g., CNN + skip connections → ResNet, Transformer + spatial embedding → ViT, Diffusion + Transformer → DiT, and windowing + Transformer → Swin Transformer. The originality often resides in the insight behind combining known elements to solve a new or particularly challenging problem, which MODEM exemplifies.
> > >
> > > The effectiveness of this approach is validated by the experimental results. The novelty and effectiveness of our idea was also positively recognized by other reviewers (i.e.,  QaHQ, 2iWh, sL85).
> > >
> > > **On Comparisons:**
> > >
> > > As we clarified in the rebuttal, our work focuses on the **unified all-weather setting**, a specific benchmark for which many recent task-specific methods were not designed or evaluated, hence their initial exclusion. When requested, we made a significant effort to provide **new comparisons against** **7** **recent SOTA methods**, which are reported in our rebuttal and summarized again below for your convenience.
> > >
> > > | Method | Snow100K-L | Snow100K-S | Snow100K-2000 |
> > > | ------ | ---------- | ---------- | ------------- |
> > > | ConvIR | 32.11      | 37.98      | 33.92         |
> > > | FSNet  | 31.62      | 37.42      | 33.76         |
> > > | MODEM  | **32.52**  | **38.08**  | **35.76**     |
> > >
> > > | Method | RainDrop  |
> > > | ------ | --------- |
> > > | AST    | 30.57     |
> > > | MODEM  | **33.01** |
> > >
> > > | Method    | Snow100K-L | Snow100K-S | Outdoor   | Raindrop  |
> > > | --------- | ---------- | ---------- | --------- | --------- |
> > > | MambaIR   | 28.59      | 33.34      | 24.73     | 32.16     |
> > > | FreqMamba | 27.09      | 33.52      | 19.89     | 30.60     |
> > > | VmambaIR  | 31.07      | 36.35      | 24.23     | 32.18     |
> > > | CODE      | 30.46      | 36.29      | 26.65     | 32.64     |
> > > | MODEM     | **32.52**  | **38.08**  | **33.10** | **33.01** |
> > >
> > > The results clearly demonstrate that our method achieves state-of-the-art performance compared to these recent methods, both on the directly overlapping tasks and on the tasks where we made our best effort to retrain their models.
> > >
> > > Furthermore, these new comparisons also highlight the superiority of our degradation estimation approach. It is crucial to note that the other methods in the tables above were trained **separately for each individual task**, whereas our unified model was trained **on all tasks simultaneously**. The fact that our unified model still outperforms these task-specific approaches is strong evidence of its effectiveness. This is particularly evident on the **Outdoor benchmark**, a challenging joint deraining and dehazing task. As the results show, methods without degradation estimation struggle significantly on this complex, mixed-degradation scenario, which underscores the power and necessity of our proposed mechanism.
> > >
> > > We hope these clarifications help resolve the concerns raised and better reflect the contributions and rigor of our work. Thank you again for your time.

---

### Official Review · Reviewer_sL85 · 2025-06-25

**Clarity:** 1
**Significance:** 3
**Originality:** 2
**Rating:** 4
**Confidence:** 4

**Summary:**

The authors propose MODEM, SSM-based image restoration model for adverse weather removal.
Beside general scanning methods, they utilize Morton scanning to better preserve spatial locality and capture contextual information.
To maximize the effect of Morton scanning, they introduce MOS2D module and training strategies.
With those contributions, their model achieves a significant margin over previous models.

**Questions:**

1. Please include more comprehensive ablation studies to validate the effectiveness of DSAM in various settings, such as presenting results when only DDEM and DAFM are used. Additionally, ablations on the loss terms should be conducted. Specifically, it is important to report the model’s performance when knowledge distillation (KD) is removed (i.e., after stage 1), and also to show how well the model performs when trained without ground truth supervision.

2. Visualization results under diverse real-world weather conditions should be provided. Datasets such as RESIDE, NTURain, and RainDS offer a range of real-world adverse weather images that can be used to demonstrate the model’s effectiveness in practical scenarios.

3. Please provide a more detailed explanation of "HistoFormer+".

4. Please show the results under different weather conditions in Fig. 3. (rain/snow)

5. The authors should also demonstrate the performance of MODEM on other adverse weather datasets, such as WeatherStream dataset, to validate its generalizability.

**Ethical Concerns:**

["NO or VERY MINOR ethics concerns only"]

**Final Justification:**

By reading the rebuttal, the concerns raised in the initial review have been resolved. The connection between the proposed model and the corresponding figures has also become clearer. In addition, the authors provided sufficient experiments and analysis regarding the loss-related question, which confirmed that the proposed approach is well-structured. Therefore, I believe the paper now presents sufficient evidence to support the proposed model, and I am increasing my score to Borderline Accept.

**Limitations:**

yes

**Paper Formatting Concerns:**

no issue i found

**Quality:**

3

**Strengths And Weaknesses:**

Strengths
1. Although the number of parameters slightly increases, it achieves faster inference speed and better performance compared to the existing state-of-the-art.
2. MODEM achieves substantial performance improvement on the Allweather dataset.
3. A structure well-suited for adverse weather removal by separating and processing long-range and localized information based on SSM.

Weaknesses
1. The explanation of the architecture is not clear. The relationship between Fig. 4 and Fig. 5 is not clearly articulated in conjunction with the text.
2. The ablation analysis is insufficient. To convincingly demonstrate the effectiveness of the proposed module, a more detailed ablation study is necessary. This should also be extended to the loss terms.
3. Lack of visualizations for real world scenarios. (e.g. fog, rain)

---

> ### Author Rebuttal · Authors · 2025-07-31
>
> **Q1: The explanation of the architecture? & The relationship between Fig. 4 and Fig. 5? (W1)**
>
> **A1:** Thank you for this question. As we described in the caption of Fig. 5, the relationship between Fig.4 and Fig. 5 is that **Figure 5 provides a detailed, "zoomed-in" illustration of the internal workings of the MOS2D** module (Fig. 4d), which is the central component of the MDSL block shown in Figure 4(c).
>
> **Figure 4 illustrates the overall framework**, as described at the beginning of Section 4. It shows that the input degraded image is processed by two parallel streams: the Main Restoration Backbone (Fig. 4a), which is composed of multiple MDSL blocks (Fig. 4c), and the Dual Degradation Estimation Module (DDEM) (Fig. 4b). The DDEM's sole purpose is to analyze input degradation and produce two "guidance" signals (Eq. 4): a global degradation descriptor ($Z_0$) and a spatially adaptive degradation kernel ($Z_1$). These two priors are then fed into the MDSL blocks (Eq. 5) of the main backbone to guide the restoration.
>
> **Figure 5 then details precisely how this guidance (**$Z_0$ **and** $Z_1$**) is used within each MOS2D module inside an MDSL block.** It shows the internal mechanisms where the feature map entering the MOS2D module is first modulated by the global prior $Z_0$ through the Degradation-Adaptive Feature Modulation (DAFM) (Eq. 7). Subsequently, the adaptive kernel $Z_1$ is used by the Degradation-Selective Attention Modulation (DSAM) (Eq. 8) to further refine the features, making the process sensitive to local degradation patterns. Finally, these modulated features are used to dynamically generate the parameters ($\Delta, B, C$) for the core S6 block (Eq. 9, Eq. 10).
>
> We will clarify this architectural relationship more explicitly in the revision.
>
> **Q2: The ablation analysis? (W2 & Q1)**
>
> **A2:** Thanks for this question. Actually, most of the required ablations had been included in the submission. We provided ablation experiments in the **third and fourth rows of Table 5** where DDEM and DAFM exist alone, and where DDEM and DSAM exist alone, respectively. **The 4th row of Table 5 shows the performance when only DDEM and DSAM are used (w/o DAFM), while the 3rd row shows the performance when only DDEM and DAFM are used (w/o DSAM).** These experiments, when compared to the full model, demonstrate that both DAFM and DSAM contribute positively to the final performance, and their combination yields the best results.
>
> The second row of Table 5 is, in fact, the experiment where the KD Loss is removed. Our KD process, driven by the KL divergence loss, is the sole mechanism used in Stage 2 to train the practical DDEM module. **Removing this KD loss means the DDEM in stage 2 has no supervisory signal and cannot learn to estimate the degradation priors.** Consequently, both DAFM and DSAM become non-functional. This scenario, which reduces our model to a standard Mamba-style backbone without our degradation-aware guidance, is precisely what is evaluated in the second row ("Morton ✓, DDEM ✗, DAFM N/A, DSAM N/A"). The significant performance drop observed in this setting powerfully underscores the critical importance of KD loss for our framework's success. We will clarify this in the revision.
>
> As for the correlation ablation suggested by the reviewer, we conducted an ablation study on the contribution of  the Pearson correlation loss. We have also performed an ablation study on the placement (before the average pooling) of the KL divergence loss. The results are presented in the table below. They confirm that each component contributes positively and that our original design. We will include these new findings in the revised version.
>
> | Setup                | Snow100K-L | Snow100K-S | Outdoor   | Raindrop  |
> | -------------------- | ---------- | ---------- | --------- | --------- |
> | w/o Correlation Loss | 32.30      | 37.79      | 32.91     | 32.69     |
> | w/o KL Loss          | 32.12      | 37.61      | 32.37     | 32.38     |
> | Replace KL Loss      | 31.85      | 37.16      | 32.77     | 32.57     |
> | MODEM                | **32.52**  | **38.08**  | **33.10** | **33.01** |
>
> **Q3: More visualization results under diverse real-world weather conditions? (W3 & Q2 & Q5)**
>
> **A3:** Thank you for this valuable suggestion. We have provided qualitative results for real-world snow removal in **Figure 6** of the main paper, as well as for real-world raindrop removal in **Table 1, Figure 8, and Figure 13**. **Due to the rules of the rebuttal process, we are unable to upload new qualitative visualizations at this stage.** To provide quantitative evidence of our model's performance on these challenging real-world scenarios, we have conducted a new set of evaluations using the non-reference image quality assessment (IQA) metric **Q-Align**. This metric is well-suited for assessing perceptual quality without needing a ground truth image. To ensure a fair and reproducible comparison, all scores were computed using the `pyiqa` library with the same default settings applied for all methods. We evaluated our model on the real-world snow dataset as well as the real-world data from the **Snow100K-Real, RainDrop, RESIDE, NTURain and WeatherStream** datasets. The results are presented in the table below.
>
> | Method      | Snow100K-Real | RainDrop   | NTURain    | RESIDE     | WeatherStream |
> | ----------- | ------------- | ---------- | ---------- | ---------- | ------------- |
> | WeatherDiff | 3.4531        | 4.0000     | 3.2031     | **3.4219** | *1.9561*      |
> | Histoformer | *3.7207*      | *4.0156*   | *3.2266*   | 3.2891     | 1.9434        |
> | MODEM       | **3.7324**    | **4.0664** | **3.2891** | *3.3164*   | **1.9863**    |
>
> As the results show, our method achieves state-of-the-art performance on real-world datasets when compared to the previous state-of-the-art method, Histoformer, and the diffusion-based approach, WeatherDiff. We will incorporate both these new **quantitative IQA results** and the promised **qualitative visual results** into the revised version of the paper.
>
> **Q4: A more detailed explanation of "HistoFormer+"? (Q3)**
>
> **A4:** Thank you for this question. "Histoformer+" refers to results from a specific pre-trained weight file, `net_g_real.pth`, provided in the official Histoformer repository, while "Histoformer" refers to the official version reported in their paper. We would like to state clearly that **all test results and model weights reported in our paper are taken directly from official papers or repositories.**
>
> The network architecture is identical to the one in their paper. However, **the authors did not provide training details for this specific** **checkpoint**. We will clarify this naming convention in the revision paper to ensure full transparency.
>
> **Q5: The results under different weather conditions in Fig. 3? (Q4)**
>
> **A5:** Thank you for this suggestion. **Due to the rules of the rebuttal process, we are unable to upload new images at this stage.** We alternatively describe these results and promise to include them in the revision.
>
> We have performed these visualizations internally for other weather conditions. They were not all placed in the submission due to space constraints. **We can observe that the feature maps for $CAh_{k-1}$ and $CBx_k$ always exhibit a clear and consistent separation between long-range and local information, regardless of the weather type.**
>
> As described in our analysis in Section 3, **the** $CAh_{k-1}$ **term consistently focuses on the broader, more diffuse degradation context (like widespread haze or dense, distributed snowflakes), while the** $CBx_k$ **term focuses on fine-grained local details, including both sharp artifacts and underlying image textures.** We will add these new visual examples of different weather conditions to the revision of the paper to provide a more thorough demonstration. Thank you again for helping us improve the paper.

---

> > ### Comment · Reviewer_sL85 · 2025-08-05
> >
> > Thank you for the rebuttal in response to my questions. The authors’ clarifications have addressed most of the concerns I raised. I trust that the points discussed will be properly reflected in the final version of the paper, and I am raising my score accordingly.

---

> > > ### Author Response · Authors · 2025-08-05
> > > **Reply to Reviewer sL85**
> > >
> > > Thank you for your valuable feedback and for your positive assessment. We agree that your suggestions have been very beneficial for improving our paper. We will ensure that all of the discussed details are included in the final revised version.

---

### Official Review · Reviewer_QaHQ · 2025-06-30

**Clarity:** 3
**Significance:** 2
**Originality:** 3
**Rating:** 4
**Confidence:** 4

**Summary:**

This paper primarily introduces an ordering mechanism based on the Morton-order for images. The proposed Morton-Order 2D Selective Scan Module (MOS2D) combines the Morton ordering with S6 blocks (Selective State Space Model). The paper argues that the proposed ordering scheme better captures the spatial locations of pixels when transforming to the 1D sequence which is fed to the S6 block. The end application is restoring images afflicted by adverse weather such as rain, haze and snow.

**Questions:**

Other than the questions listed in weaknesses, a few minor questions are:

**Q1.** How is Fig. 7 generated? Are the features taken from the latent or are they priors?

**Q2.** After pixel at (0,0) and (0,1) are scanned, the next is at (1,0) and not (0,2), making a Z curve along this specific path. Have the authors considered combining multiple scan paths?

**Q3.** Citation of the original Morton order paper is missing [1].

[1] Morton, Guy M. "A computer oriented geodetic data base and a new technique in file sequencing." (1966).

**Comments**

Ordering schemes like Morton-order have been explored in the literature in context of point clouds [1] and now even for image restoration [2], the authors should consider citing these works. Although the latter is fairly recent and can be considered contemporary to this work.

[1] Chen, Guangyan, et al. "Pointgpt: Auto-regressively generative pre-training from point clouds." Advances in Neural Information Processing Systems 36 (2023): 29667-29679.

[2] Ouyang, Zongzhi, and Wenhui Li. "MMamba: Enhancing image deraining with Morton curve-driven locality learning." Neurocomputing 638 (2025): 130161.

**Ethical Concerns:**

["NO or VERY MINOR ethics concerns only"]

**Final Justification:**

My major concerns were regarding comparison with other scanning approaches in Mamba style architectures. The authors provided detailed comparisons with several different Mamba-style methods, motivating that Morton-ordering is computationally efficient and achieves better performance. The new details alleviated my concerns and I have raised the rating to be positive.

**Limitations:**

Yes, the authors discuss the limitations of their work, but no discussion on the societal impact is provided.

**Quality:**

3

**Strengths And Weaknesses:**

**Strengths**

**S1.** The paper is well-written and straight-forward to follow.

**S2.** The proposed method scores strong results on the adverse weather dataset, and performs efficiently (although measured on 256x256 resolution input images).

**Weaknesses**

**W1.** One of the core claims of this work is that the proposed ordering scheme is better at preserving spatial locations when transforming 2D images to 1D sequence. However, there is no comparison with a single Mamba-style restoration network in Tabs. 1-4. There exist many works that explore how well SSMs perform in image restoration problems, many of which the paper already cites in the related work, but does not compare to. This is crucial to understand if indeed the proposed ordering mechanism is better than the prior mechanisms such as OSS in [1], or SS2D in [2], etc.

[1] Shi, Yuan, et al. "Vmambair: Visual state space model for image restoration." IEEE Transactions on Circuits and Systems for Video Technology (2025).

[2] Liu, Yue, et al. "Vmamba: Visual state space model." Advances in neural information processing systems 37 (2024): 103031-103063.

**W2.** Have the authors thought about discontinuities that exist in the scanning order? In Fig. 1, this is visible in (d) when once the first 4 columns and first 4 rows are complete, the next one is col 5, row 1 pixel which is not spatially next to the last pixel that was scanned. Due to this, it is necessary to understand why the proposed scheme is better and why zigzag like scheme [1] or Hilbert curve based scanning [2] is not better?

[1] Hu, Vincent Tao, et al. "Zigma: A dit-style zigzag mamba diffusion model." European Conference on Computer Vision. Cham: Springer Nature Switzerland, 2024.

[2] Jeong, Min Wu, and Chae Eun Rhee. "LC-Mamba: Local and Continuous Mamba with Shifted Windows for Frame Interpolation." Proceedings of the Computer Vision and Pattern Recognition Conference. 2025.

**W3.** Since the training setup is a 2-stage pipeline, I assume the entire model (along with DDEM) is trained given the loss in eq.11. If so, during stage 2, does the restoration backbone now start from scratch or the stage 1 acts as pretraining for the restoration method? In the latter stage (not starting from scratch), there is an issue with the GT information leaking as priors $Z_{0}$ and $Z_{1}$ into the main restoration network.

---

> ### Author Rebuttal · Authors · 2025-07-31
>
> **Q1: Measured on 256x256 resolution? (S2)**
>
> **A1:** Thank you for your positive feedback. To be more comprehensive, we have now evaluated the inference speed on larger resolutions. The results (in ms), tested on a single RTX 3090 GPU, are shown below:
>
> | Input Size | WGWSNet | WeatherDiff | Histoformer | MODEM   |
> | ---------- | ------- | ----------- | ----------- | ------- |
> | 256*256    | 24.83   | 1.67e6      | 109.07      | 92.86   |
> | 512*512    | 110.34  | 5.37e6      | 576.15      | 443.02  |
> | 1024*1024  | 439.13  | 1.35e7      | 3056.29     | 1946.34 |
> | Avg. PSNR  | 31.54   | 31.57       | 33.68       | 34.18   |
>
> The average PSNR is calculated across the 4 benchmarks shown in Table 1 of the main paper. We will include this more comprehensive evaluation in the revision.
>
> **Q2: Compare with Mamba-style network? (W1)**
>
> **A2:** Thank you for your question. To our knowledge, MODEM is the first Mamba-style architecture for the unified all-weather task, which is why other Mamba-style methods were not in our initial comparisons.
>
> As suggested, we further retrained three prominent Mamba-style restoration networks, separately on each of the three tasks within the all-weather setting. As these models were not specifically designed to our target task, we tried our best to retrain them using configurations provided by the authors and following the data usage that is commonly-adopted by the community (e.g. Histoformer and ours).
>
> | Method    | Snow100K-L | Snow100K-S | Outdoor   | Raindrop  |
> | --------- | ---------- | ---------- | --------- | --------- |
> | MambaIR   | 28.59      | 33.34      | 24.73     | 32.16     |
> | FreqMamba | 27.09      | 33.52      | 19.89     | 30.60     |
> | VmambaIR  | 31.07      | 36.35      | 24.23     | 32.18     |
> | MODEM     | **32.52**  | **38.08**  | **33.10** | **33.01** |
>
> With respect to the scanning style, we have already provided a comparison in ablation study. As shown in **Table 5 (row 1)**, replacing our Morton-order scan with a raster scan (as used in MambaIR) leads to a performance drop. This reveals the benefit of Morton-order's locality preservation. Additionally, to further investigate the impact of the scanning scheme itself, we here offer an expanded ablation study **within our** **MODEM****,** comparing the Morton scan against others. As can be seen from the results, our Morton manner achieves the best performance.
>
> | Method               | Snow100K-L | Snow100K-S | Outdoor   | Raindrop  |
> | -------------------- | ---------- | ---------- | --------- | --------- |
> | Raster (e.g.MambaIR, Vmamba) | 32.33      | 38.03      | 32.89     | 32.69     |
> | Continuous (zigzag)  | 32.17      | 37.74      | 32.35     | 32.59     |
> | OSS (VmambaIR)       | 32.14      | 37.39      | 32.61     | 32.11     |
> | Local (LocalMamba)   | 32.23      | 37.75      | 32.60     | 32.53     |
> | Hilbert (LC-Mamba)   | 32.46      | 37.96      | 32.99     | 32.82     |
> | Morton (Ours)        | **32.52**  | **38.08**  | **33.10** | **33.01** |
>
> We will include this additional comparison in the revision for completeness.
>
> **Q3: Discontinuities of Morton-order scan? Combine multiple scan paths? (W2 & Q2)**
>
> **A3:** Thank you for this question. Since any sequential 2D scan will inevitably suffer from internal discontinuities, the key is to find a scanning method that is better suited for the image restoration task. As prior research has demonstrated [1, 2], for images, **immediately surrounding pixels are** **typically** **far more informative than pixels in the next row or column**. The Morton-order scans an image by processing it in small, contiguous block-like regions, moving from one adjacent block to the next **(refer to the scale-space theory)**. This ensures that **pixels that are close in the 2D image stay close in the 1D sequence,** unlike a raster or zigzag scan (continuous scan in Fig. 1b), which scans line-by-line and creates large gaps between long-distance pixels.
>
> [1] Regional Attention for Shadow Removal, ACM MM 2024
>
> [2] A Comparative Study of Image Restoration Networks for General Backbone Network Design, ECCV 2024
>
> While LocalMamba (Fig. 1c) also attempts to solve this issue, it is limited to applying a raster scan within pre-defined local windows and then using another raster scan to connect them, which still does not ensure pixel-level proximity across the image, although mitigated to some extent. As for Hilbert curve, it is another locality-preserving alternative. However, it comes with a significantly higher computational cost, whereas the **Morton-order can be calculated efficiently with simple bitwise operations as shown in our Eq. 6**. To be clear, we compared the inference speed of the Morton and Hilbert scans. For the Hilbert scan, we used the official implementation from LC-Mamba. The speed (ms) of different resolutions are shown in the table below, with all tests performed on a single RTX 3090 GPU. Our method is significantly faster than Hilbert.
>
> | Input Size   | 256*256   | 512*512    | 1024*1024   |
> | ------------ | --------- | ---------- | ----------- |
> | Hilbert scan | 604       | 10134.91   | 88288.46    |
> | Morton scan  | **92.86** | **443.02** | **1946.34** |
>
> **To better validate these arguments, we have conducted a new ablation study of scan patterns, with the results shown in Answer 2.**
>
> **We only need a single scan.** This evidence confirms that a **single** Morton-order scan is both effective and efficient. Therefore, for architectural simplicity, we prioritize a superior single-pass approach rather than compensating for a weaker scan with multiple paths. We will clarify this in more detail in the revision.
>
> **Q4:** **GT** **leaking? (W3)**
>
> **A4:** Thank you for this question. The parameters for the Stage 2 model are initialized from the weights trained in Stage 1 as mentioned in Appendix A.1. First, and most importantly, **at no point during training was our model ever exposed to any images from the test sets, while** **at inference,** **our** **trained** **model** **merely** **takes** **the degraded image** **as input**. This is how supervised deep learning works, and comparison fairness is guaranteed in this work.
>
> We tried to capture the point from the reviewer on this question. There might be some confusion about our two-stage training. Thus, to make it clear, we here emphasize the pipeline of the two-stage strategy:
>
> In Stage 1, the input to the DDEM consists of both the GT and the degraded images from the training set, while the main backbone only receives the degraded image. The DDEM ultimately outputs only a single degradation vector. By providing the DDEM with both the degraded image and GT from the training set, the DDEM is explicitly trained to **understand the difference between the degraded image and GT**, and thereby learn to extract the degradation representation required for the main restoration backbone, which serves as the ideal supervisory target for Stage 2.
>
> In Stage 2, the inputs to both the DDEM and the main backbone **consist of only the degraded image, without any GT**. The Stage 2 model inherits its parameters from Stage 1, and **the DDEM in this stage receives only the degraded image as input**. Simultaneously, there is a frozen DDEM receiving both the GT and the degraded image. **The degradation information from this frozen DDEM is then used to supervise the trainable DDEM via the KL** **divergence** **loss, as shown in Eq. 13.** **So, the paired GT-degraded images from the** **training set** **are used to better train the degradation estimation module, but this is entirely separate from GT leakage during inference.**
>
> Furthermore, **our** **MODEM** **demonstrates excellent generalization capabilities benefiting from our two-stage degradation estimation strategy.** As shown in Figure 6, MODEM achieves superior results on real-world images. **This is a dataset** **that** **was not explicitly trained or finetuned on and for which no GT was ever available**. To quantify this, we computed non-reference metrics (Q-Align, MUSIQ), and our method achieves the best results, as shown below. This strong real-world performance is compelling evidence of a robust, generalizable representation, not overfitting via information leakage.
>
> | Method      | Q-Align$\uparrow$ | MUSIQ$\uparrow$ |
> | ----------- | ------------------- | ----------------- |
> | MODEM       | **3.5586**          | **59.6042**       |
> | Histoformer | 3.5449              | 59.4040           |
> | WeatherDiff | 3.4512              | 59.4493           |
>
> **Q5: Explanation of Figure 7? (Q1)**
>
> **A5:** Thank you for this question. **Since Histoformer does not have a dedicated prior design, extracting features from the latent space is the only comparable approach.** So the features used to generate the t-SNE plot in Figure 7 were extracted from the latent. The relative position of feature extraction within our network was kept consistent with that of Histoformer. We will add a precise description of this process to the revision of the paper for clarity.
>
> **Q6: Reference? (Q3 & Comments)**
>
> **A6:** Thank you for providing the reference to the Morton paper[1] and its application in the 3D vision domain[2]. We will certainly cite these works in our revision to better contextualize our approach and strengthen the discussion. Regarding MMamba[3], as you noted, this work was published after our submission deadline. Although the scanning mechanisms are similar, our work's primary focus lie in the **modeling and perception of degradation estimation**, a concept not central to MMamba. We will be happy to cite it in the revision.
>
> [1] A computer oriented geodetic data base and a new technique in file sequencing.
>
> [2] Pointgpt: Auto-regressively generative pre-training from point clouds.
>
> [3] MMamba: Enhancing image deraining with Morton curve-driven locality learning.

---

> > ### Comment · Reviewer_QaHQ · 2025-08-04
> >
> > I thank the authors for their rebuttal and for providing extra details regarding their method. My concerns have been sufficiently addressed, and as such I have raised the score.
> > I encourage the authors to include these extra details in the final version so that the reader can thoroughly understand the proposed method.

---

> > > ### Author Response · Authors · 2025-08-04
> > > **Reply to Reviewer QaHQ**
> > >
> > > Thank you for your positive feedback and for your valuable suggestions. We will be sure to include all of these details in the revised version of our paper.

---

### Note · Authors · 2025-08-12

We sincerely thank PCs, ACs, and all reviewers for their time and effort. Specifically, we would like to highlight the opinions from the reviewers:

- **R-QaHQ** initially noted that our "proposed method scores strong results... and performs efficiently." After rebuttal, he/she confirmed the concerns were "sufficiently addressed" and raised the score.

- **R-sL85** praised our method "achieves a significant margin over previous models" with a "structure well-suited for adverse weather removal." He/she subsequently confirmed our clarifications "addressed most of the concerns" and raised the score.

- **R-2iWh** began with a positive assessment, stating our model "sounds interesting" and that "Using Morton... brings new insight to this community" . After our rebuttal, he/she reiterated that the "idea of this paper itself is appreciated."

The reviewers also provided valuable suggestions for additional comparisons with specific one-task models, although we have compared our MODEM with SOTA unified weather restoration methods and one-task models following previous arts. In response to the suggestions, **we tried our best to perform these experiments during rebuttal, and each of them confirmed that the concerns were addressed.**

However, we respectfully point out a disconnect with the assessment from **R-6wM5**. The reviewer's final remark that our comparisons are **"out of date... although the authors add new experiments"** is confusing, as it seems to dismiss our experimental efforts, **including new comparisons against 7 recent SOTA methods.**

Furthermore, the R-6wM5's **reversal** on novelty is particularly concerning. **He/She initially said our method was "interesting" and "effective"** but later claimed it lacked novelty without specific justification, contradicting his/her own initial view and the consensus from other reviewers. We have repeatedly emphasized our method's novelty and contributions in both the submission and rebuttal. And the superiority of our degradation estimation mechanism is clearly demonstrated in our comparisons with other unified methods (Table 1), the better-clustered t-SNE feature space (Figure 7), AND by the fact that our unified model surpasses even recent task-specific models in Table 2-4 and in our new rebuttal experiments.

**We also addressed all of the R-6wM5's other concerns on figures and ablation studies, to which no further objections were raised.**

We hope you will consider these points in your final decision.

---

### Decision · Program_Chairs · 2025-09-17

**Decision:**

Accept (poster)

**Comment:**

This paper proposes an effective MODEM for image restoration. It explores morton-coded spatial ordering with selective state-space models to model both local and non-local information.

This paper received reviews with mixed ratings. The major concerns include limited novelty and insufficient comparisons with recent SOTA methods.

Based on the provide rebuttal and discussions with reviewers, the authors solve the concerns of reviewers, especially for comparisons with suggested methods by reviewers.

Based on the rebuttal and recommendations of reviewers, the paper can be accepted. The authors are suggested to revise the paper according to comments accordingly.